# Seismic and gravity constraints on plate flexure and mantle rheology along the whole Hawaiian-Emperor seamount chain

A. B. Watts [1,7] ✉, C. Xu [1,2,7] ✉, P. Wessel[6], D. J. Shillington [3], B. Boston [4] & R. Dunn [5]

The Hawaiian-Emperor seamount chain is a hotspot-generated intraplate volcanic chain that has yielded key constraints on the rigidity of Earth's tectonic plates. However, previous studies have shown significant variability in the effective elastic thickness, $T_e$, a proxy for the long-term strength of the lithosphere, along the chain. While low $T_e$ (10-20 km) at the Emperor Seamounts and high $T_e$ (17-37 km) at the Hawaiian Ridge are expected because of their differences in volcano and plate age, the change between them appears abrupt and occurs in the vicinity of the bend in the chain. To investigate this variability, we estimated $T_e$ along 3000 profiles - spaced 2 km apart along the chain - from gravity and flexure modeling, calibrated using constraints from deep seismic experiments carried out in 2018 and 2019. Here we show that, contrary to previous predictions, $T_e$ changes gradually along the chain and that weak zones can exist within the interior of a large, otherwise rigid, plate and may, we speculate, facilitate the initiation of intra-oceanic subduction in response, for example, to changes in plate motion and eventually lead to their break-up.

The Hawaiian-Emperor seamount chain (HESC) is a pre-eminent example of an intraplate hotspot-generated volcanic chain that has provided key constraints on the finite rotations of Earth's tectonic plates as well as their long-term flexural rigidity. The pioneering studies of Vening Meinesz[1] and Gunn[2], for example, showed using gravity anomalies that the southeast end of the chain was regionally rather than locally isostatically compensated. Subsequent studies[3,4] demonstrated the gravity anomalies could be explained by simple models in which the volcanoes that comprise the chain loaded a thin elastic or viscoelastic plate (i.e., a plate with a thickness that is small compared to the radius of curvature of flexure) with uniform thickness. Watts[5] found differences along the chain with small values (10-20 km) of the effective elastic thickness, $T_e$, a proxy for the long-term strength of the lithosphere, at the Emperor Seamounts and large values (17–37 km) at

the Hawaiian Ridge. Since the chain is underlain mostly by Cretaceous Normal Polarity oceanic crust and the Emperor Seamounts are significantly older than the Hawaiian Ridge, Watts[5] proposed that $T_e$ was a function of the age of the Pacific plate at the time of loading.

While subsequent forward modeling studies[6–11] have generally confirmed these results, they reveal an abrupt change in $T_e$ at the HESC bend (~48 Ma[12]), which is not expected from the volcano load (0-80 Ma[12]) and plate (83-121 Ma[13]) ages (Supplementary Fig. S1). Spectral studies based on the moving window admittance[14] and wavelet[15] techniques reveal more gradual changes at the bend. But the windows used by Kalnins and Watts[14] are hundreds of km in width and so tend to smear out spatial variations in $T_e$. Wavelet techniques are spatially localized, but the $T_e$ values resolved by Lu et al.[15] are relatively high (~35 km) at the Emperor Seamounts and relatively low (~25 km) at

[1]Department of Earth Sciences, University of Oxford, Oxford, UK. [2]Key Lab of Submarine Geosciences and Prospecting Techniques, Ministry of Education, and College of Marine Geosciences, Ocean University of China, Qingdao, China. [3]School of Earth and Sustainability, Northern Arizona University, Flagstaff, AZ, USA. [4]Auburn University, Department of Geosciences, Auburn, AL, USA. [5]Department of Earth Sciences, School of Ocean and Earth Science and Technology, University of Hawaii at Manoa, Honolulu, HI, USA. [6]Deceased: P. Wessel. [7]These authors contributed equally: A. B. Watts, C. Xu. ✉e-mail: tony.watts@earth.ox.ac.uk; chong.xu@earth.ox.ac.uk

the western Hawaiian Ridge, which are difficult to reconcile with previous results, including those of Kalnins and Watts[14]. Irrespective, previous forward modeling estimates of $T_e$ beneath the HESC are consistently lower than those determined by previous workers for the same age of plate presently being subducted at the Circum-Pacific Trench-Outer Rise (CPTOR) (Supplementary Fig. S1).

Comparisons of observed and calculated $T_e$ based on Yield Strength Envelope (YSE) profiles (referred to here as the 'yield $T_e$') show that the brittle law of Byerlee[16] and the ductile flow laws of Goetze[17,18], as well as other Low Temperature Plasticity (LTP) laws[19,20] describe well the observed $T_e$ deduced by Hunter and Watts[21] in their inversion A1 at the CPTOR. These same laboratory-based laws, however, appear too strong at both the Emperor Seamounts[22] and the Hawaiian Ridge[23]. There has therefore been a debate[24] as to why the strength of the Pacific oceanic plate might differ between its interior and its boundary, and the role played by factors such as the timescales of loading, spatial differences in brittle and flow law parameters, or thermal rejuvenation and melt-assisted flexure.

The best observational constraint on flexure driven by volcanic loads is seismic data, which, unlike gravity or geoid data, have the potential to image the surfaces of flexure directly. Unfortunately, seismic constraints along the HESC are limited to a few single reflection and refraction transects. The earliest experiments[25,26] used shots at sea and stations on land and revealed relatively thick oceanic crust (~15– ≥21 km) beneath Maui and Hawaii and relatively thin crust (~6–7.5 km) in flanking regions but no details of the transition between them. The first multichannel seismic experiment specifically designed to test flexure was carried out onboard R/V *Robert D. Conrad* and R/V *Kana Keoki* in 1982 along 3 transects of the Hawaiian Ridge in the vicinity of Oahu[27]. The reflection data, which have recently been reprocessed[28], continuously imaged the top of oceanic crust and Moho in the vicinity of Oahu. The refraction data resolved the velocity structure at 9 Expanding Spread Profile mid-points, which was used together with the free-air gravity anomaly to constrain the $T_e$ and the flexure.

In 2018 and 2019 a multichannel seismic experiment was carried out along four 550-km-long transects of the HESC at Jimmu/Suiko guyot, Oahu/Kauai and Maui/Hawaii using a 15-km-long streamer and a 40-element air gun array onboard R/V *Marcus G. Langseth* (thick red lines, Fig. 1). The refraction data[29–32], which was acquired using 35 Ocean Bottom Seismometers, revealed the internal *P*-wave velocity structure of these seamounts and ocean islands, including the geometry of their high *P*-wave velocity (>6.5–7.0 km s$^{-1}$) and dense cores (>2710–2972 kg m$^{-3}$), the top of the pre-flexed oceanic crust in the flanking moats, and on Emperor Seamounts Line 1 and Hawaiian Ridge Lines 1 and 2 (Fig. 1) the bottom of the oceanic crust, or Moho, beneath the volcanic edifice. The reflection data[33] has revealed spatial variations in regional crustal structure and thickness variations, Moho reflective characteristics, and the locations of buried oceanic fracture zones that intersect the HESC.

In this work, we use seismic constraints on the crust and mantle structure together with 2D and 3D forward gravity and flexure modeling to evaluate $T_e$ on 3000 profiles spaced 2 km apart along a 'trail' that connects the summits along the HESC. We show here the results of the modeling and examine their implications for crustal structure, plate flexure, and the long-term rheology of the Pacific oceanic lithosphere.

## Results

The main results of our modeling are 3000 estimates of $T_e$ and a range of estimates for the densities of the root and moat infill. The degree to which the model predictions fit the observed free-air gravity anomaly data is shown in Supplementary Movies 1 and 2 and along 6 selected profiles of the HESC (Fig. 1, inset) in Fig. 2 for Model D and in Supplementary Fig. S2 for Model C. The Root Mean Square (RMS)

difference between observed and calculated gravity anomaly has a well-defined minima (e.g., Supplementary Fig. S3) and ranges from 9.2–13.6 mGal and 7.8–13.0 mGal for Models D and C, respectively, which we consider excellent in the presence of observed free-air gravity anomalies that are in the range −100 to +300 mGal.

While the RMS differences for Models C and D are similar, they yield different densities for the load and root and moat infill. Model C is associated with peak load densities in the range 2764–2840 kg m$^{-3}$ (Supplementary Fig. S4), consistent with previous estimates along the HESC. But, the root and moat infill densities for Model C are high and in the range 2450–3200 kg m$^{-3}$. Model D, in contrast, is associated with moat infill densities in the range 2500–2725 kg m$^{-3}$ (Fig. 2), which are generally consistent with previous studies along the HESC. But, the root infill densities in Model D appear low, at least for Jimmu, Ojin, Kure, and French Frigate Shoals. We explain these results as indicating that both models require some increase in mass compared to 'normal' values: Model C achieves it through increasing the density of the infill, while Model D achieves it through a boost to the 'driving' load (i.e., the surface bathymetric load above the regional bathymetry), γ. Densities for the load and infill along the entire HESC are summarized in Supplementary Fig. S4.

Significantly, Models C and D yield similar results for $T_e$ along the HESC (Supplementary Fig. S5b and Fig. 3a), the mean and RMS difference between them being 1.7 and 4.6 km, respectively. Both models reveal small $T_e$ values at the Emperor Seamounts, relatively high values at the Hawaiian Ridge, and a $T_e$ that varies gradually, not abruptly, on either side of the HEB. The increase towards the young end of the Hawaiian Ridge appears to occur independently of the age of the plate at the time of loading, which is constant at ~90 Ma, although seafloor ages are not well known in Cretaceous Normal polarity oceanic crust. Similar results were found by Watts et al.[34] from a reciprocal admittance and Kalnins and Watts[14] from a moving window admittance. In addition, the $T_e$ values are compatible with previous forward modeling results at the central Emperor Seamount chain[29] and the central Hawaiian Ridge[35].

The comparison of our results with seismic data requires that we first use the $T_e$ structure derived from the models to calculate the top and bottom (i.e., Moho) of the flexed oceanic crust. To do this, we subdivided the bathymetry along the HESC into 27 contiguous loads (Supplementary Fig. S6b) and computed the flexure associated with each load using the best-fit $T_e$ derived from gravity modeling.

Figure 3b (black lines) shows the 'upstream' flexure (i.e., the flexure in a direction opposite to that of load migration) associated with each load and the progressive flexure (thick red dotted line) obtained by summing the 'upstream' and the 'downstream' flexure (i.e., the flexure in the direction of load migration) based on the best fit $T_e$ for Model D.

We justify our choice of Model D with a 'load boost' for the Emperor Seamounts because they are mainly guyots (Fig. 3c), so have lost a substantial amount of their summit region as the once ocean island was being reduced to sea level. The Hawaiian Ridge, in contrast, comprises more atolls and islands (Fig. 3c) than the Emperor Seamounts, and so we suspect less summit removal and hence less of a 'load boost' along this part of the HESC. There are guyots on the Hawaiian Ridge, and there is evidence of large-scale landsliding, which might justify some 'load boost'. However, Supplementary Fig. S5b reveals little difference between the $T_e$ along the Hawaiian Ridge derived from Model D with a 'load boost', γ, of 1.35 (or γ = 1.15) and Model C with γ = 1.00.

Figure 3c shows the 'driving' load (light blue shaded region), the calculated root infill (intermediate blue shaded region), and the calculated flexed oceanic crust (dark blue shaded region), obtained by summing the individual seamount and ocean island loads to the progressive flexure. We assumed in the calculation a uniform thickness for the flexed crust of 5.3 km and 6.0 km at the Emperor Seamounts and

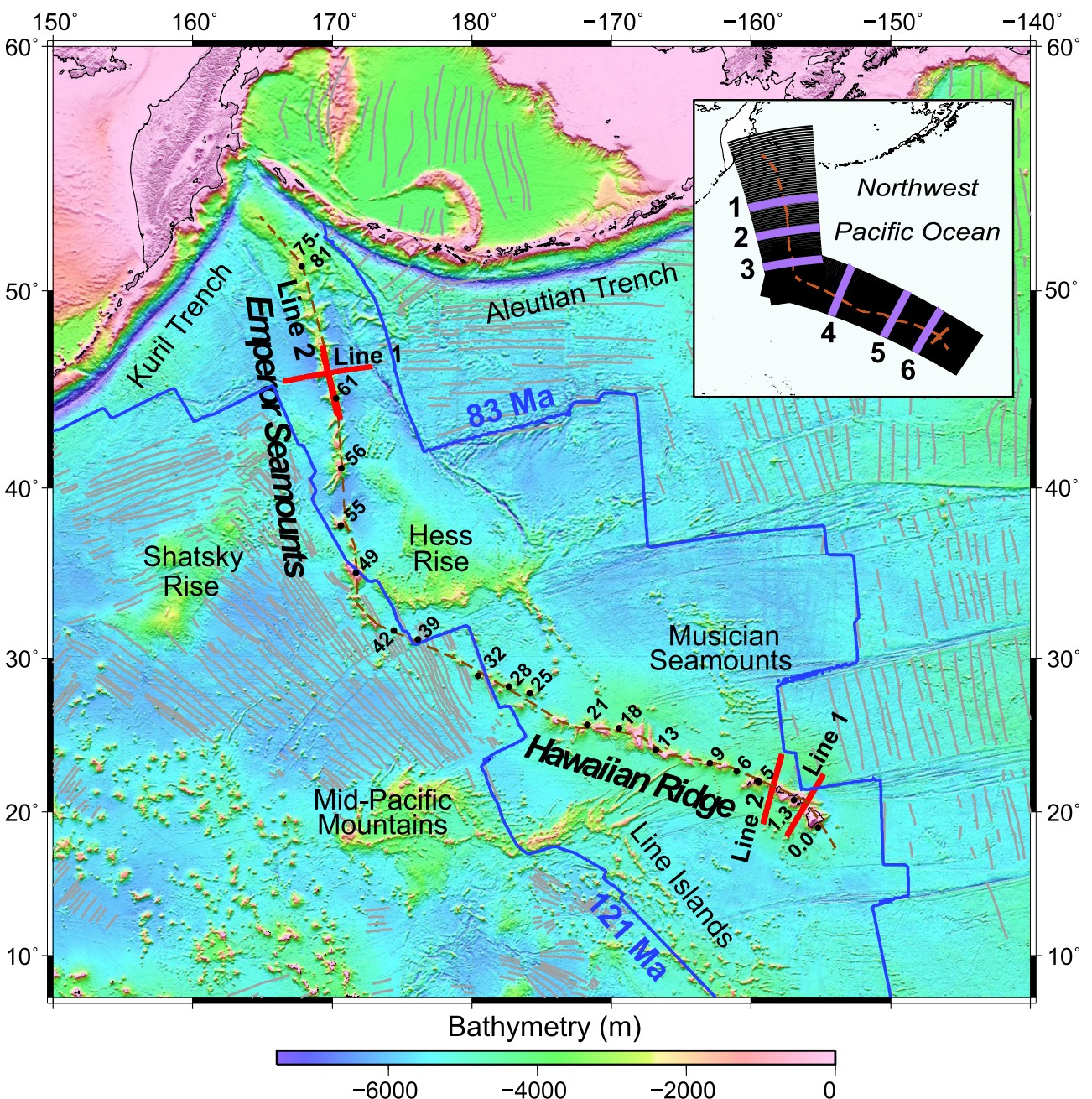

**Fig. 1 | Bathymetry map of the Northwest Pacific Ocean region based on a SRTM15 + V2.4 bathymetry grid[65], showing the location of the four transects acquired during the 2018 and 2019 seismic experiments (Lines 1 and 2 Hawaiian Ridge; Lines 1 and 2 Emperor Seamounts).** The red dashed line shows the 'trail' along which the bathymetry profile in Figs. 3 and 6 have been constructed. Grey lines show the magnetic anomaly lineations used to construct the seafloor age grid[13]. Blue lines show the 83 and 121 Ma isochrons that define the boundaries of Cretaceous Normal Polarity oceanic crust. Bold numbers with black filled circles show approximate surface sample ages in Ma[79–84]. The inset shows a subset of the profiles (at 20 km spacing for better visualization) along which the gravity anomaly has been calculated and compared to observations. Thick purple lines locate the profiles shown in Fig. 2.

Hawaiian Ridge, respectively, based on seismic refraction data[30–32]. The figure shows that oceanic crust generally shallows along the Emperor Seamounts across the HEB towards Colahan and Hancock seamounts and then deepens along the Hawaiian Ridge towards Hawaii. Moho depths vary between ~12 and 18 km, with the shallowest depth to Moho between Colahan and Hancock and the deepest depth, not beneath Hawaii in the Hawaiian Ridge, as might be expected, but beneath Koko in the Emperor Seamounts. We attribute this to the relatively large size of the Koko Seamount and its emplacement on relatively young oceanic lithosphere that is mechanically weaker on long timescales than the older lithosphere that underlies Hawaii.

Figure 4 compares the *P*-wave velocity models of Xu et al.[30], MacGregor et al.[31] and Dunn et al.[32] to an overlay of the multichannel seismic reflection data of Boston et al.[33] along each 500- to 550-km-long 'dip' transects of the HESC (Fig. 1). There is a good general agreement between the observed seismic structure and the calculated top and bottom of the oceanic crust derived from progressive flexure and gravity modeling at Emperor Seamounts Line 1 and Hawaiian Ridge Line 2 (Thick dashed white lines, Fig. 4a, b). The agreement is not as close for Hawaiian Ridge Line 1 (Fig. 4c) despite our taking into account here of the contribution of both the flexure of the Cretaceous seamounts to the southwest of Hawaii, which formed at or near a mid-

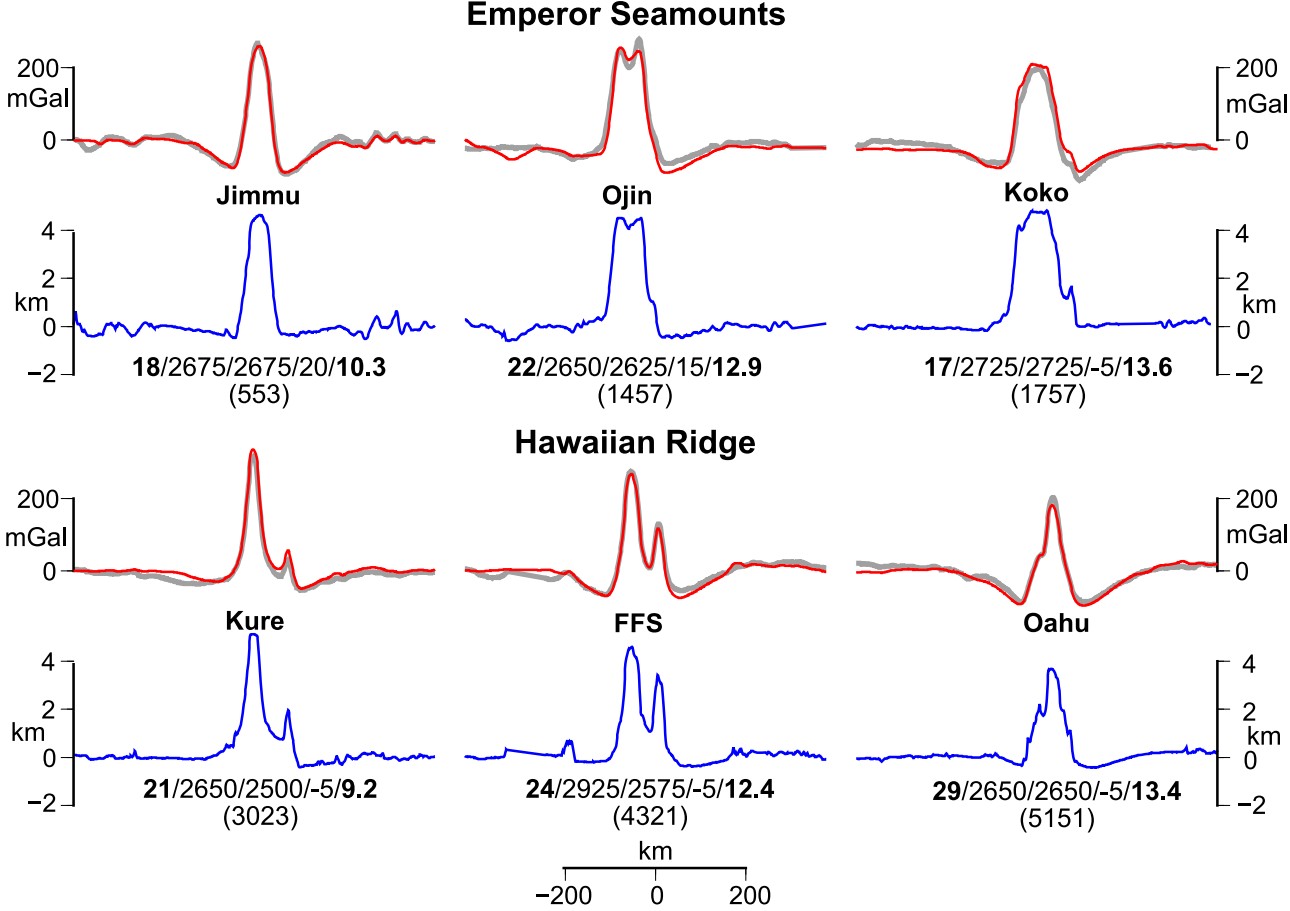

**Fig. 2 | Comparison of selected observed and calculated gravity anomalies along cross-sections of the HESC at 553, 1457, 1757, 3023, 4321, and 5151 km distance along the HESC from Detroit Seamount (Fig. 1, inset).** Grey lines show the observed free-air gravity anomaly based on Sandwell et al. V29.1[66]. Blue lines show the bathymetry. Red lines show the calculated gravity anomaly based on Model D, which includes a 'load boost' γ = 1.35. Numbers below the bathymetry profiles show the best fit values (from left to right) of $T_e$ (km, bold font), root infill density (kg m$^{-3}$), moat infill density (kg m$^{-3}$), vertical shift (mGal) applied to the calculated gravity, and the Root Mean Square (RMS) difference (mGal, bold font) between observed and calculated gravity anomaly. Plots of the RMS difference against $T_e$ reveal an uncertainty in the best fit $T_e$ of 1–2 km (Supplementary Fig. S3).

ocean ridge, and the difference in thickness of the oceanic crust, which is thinner north of Hawaii than it is to the south[31]. We consider the fit acceptable, however, given that the line transects the young end of the HESC and it is possible that isostatic equilibrium here is still not complete. For example, Hawaiian Ridge Line 1 intersects the 'donut' of earthquakes centered on Hawaii[36], which indicates active deformation and an ongoing, dynamic response to volcano loading rather than a static one, as we have assumed in the flexure modeling.

Irrespective, we believe the agreements in Fig. 4 to be close enough that we can, with some confidence, use the flexure due to the 27 contiguous loads, together with their $T_e$ structure, to estimate the volume flux of the material that comprises the 'driving' load and the root and moat infill loads. Supplementary Fig. S7 shows, for example, that the Model D $T_e$ structure overlaps previous $T_e$ results based mainly on forward modeling in the Pacific Ocean, outside of the French Polynesia region, and Supplementary Fig. S8 shows that the volume flux estimates based on this $T_e$ structure, the average density of the load and infill, and the age of the seamounts and ocean islands, generally agrees well with the previous estimates of Wessel[37]. The highest volume flux estimates are at Hawaii (-0.4 Ma) and the lowest values at Colahan and Hancock (-34–38 Ma), the main difference being the absence in our results of the peak identified by Wessel[37] at -12–14 Ma.

We have so far only considered surface loads in our modeling. There is evidence that oceanic crust underlying seamounts and ocean

islands may also have been subject to subsurface (i.e., buried) loads such as those associated with magmatic underplating[27] and intrusion[38]. The seismic data in Fig. 4 show, however, little evidence of magmatic underplating. Rather, the mantle that underlies flexed oceanic crust is remarkably homogeneous with regard to its $P$-wave velocity structure, suggesting little or no effect of large-scale underplating on either the gravity anomaly or flexure. In contrast, the seismic data reveal high velocity, dense, intrusive bodies within the core of the edifice that may act to boost the surface load. We address this here by using a 3D bathymetry-dependent load, for example, as used by Watts et al.[22], based on the seismic $P$-wave velocity structure, where the density of the load can vary laterally between a seamount flank and its dense core, rather than assume a constant density load.

## Discussion

The $T_e$ structure and flexure calculations along the HESC have implications for plate flexure and the long-term rheology of the Pacific plate. The pioneering studies of Goetze[17,18] showed, for example, how data from experimental rock mechanics could be used to predict the yield strength of the plates at pressure and temperature conditions applicable to the oceanic lithosphere. In particular, a YSE could be constructed for different thermal ages of an oceanic plate and brittle and ductile flow laws and that the observed curvatures of flexure could be used together with the YSE to calculate the 'yield $T_e$', which is

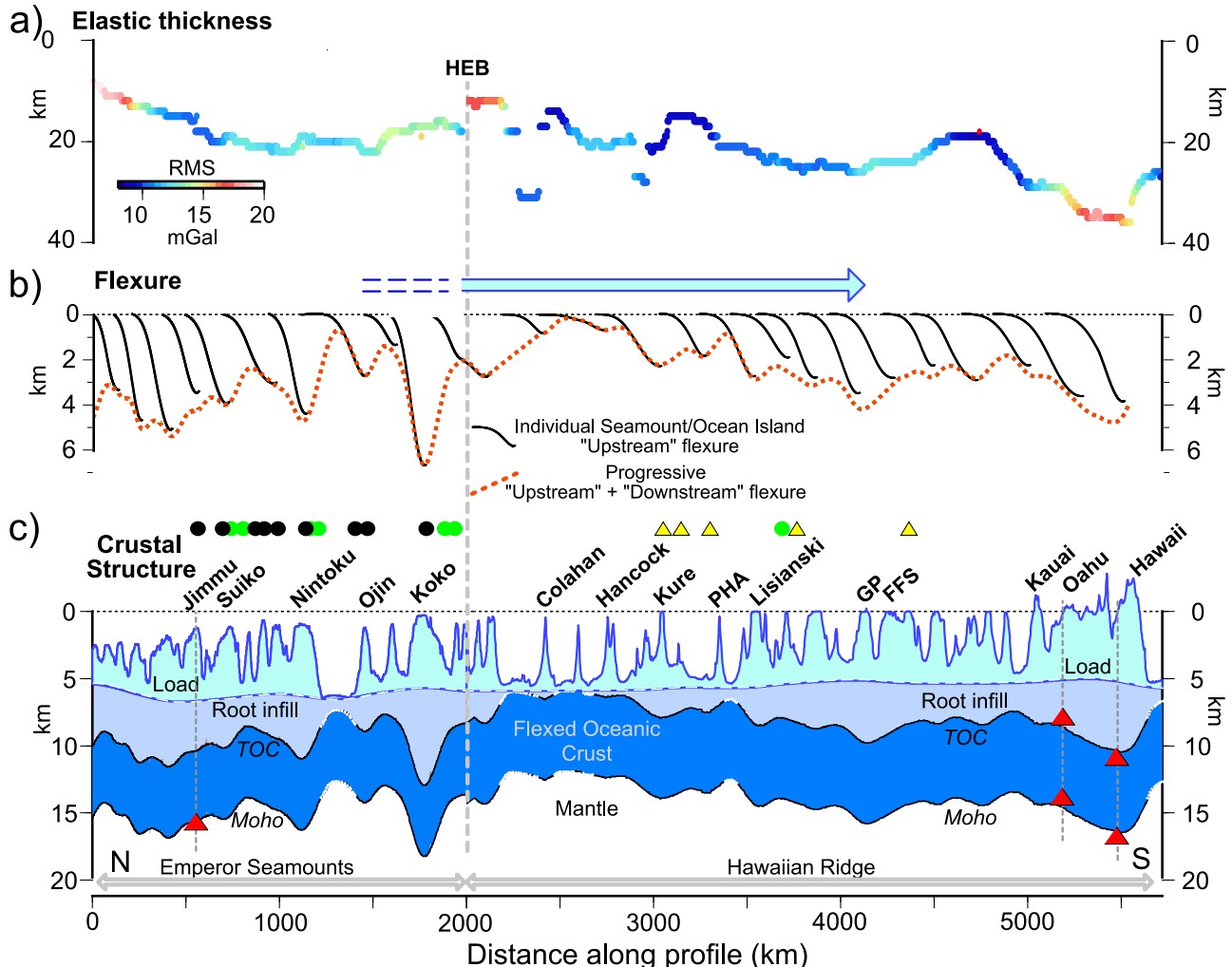

**Fig. 3 | Flexure along the HESC based on the $T_e$ derived from gravity modeling. a** The $T_e$ derived from gravity modeling. The color bar shows the RMS difference between the observed and calculated gravity anomaly. HEB = Hawaiian-Emperor Bend. **b** The 'upstream' flexure (black solid lines) associated with the individual seamount and ocean island loads. The red dotted line shows the progressive flexure due to all the loads. **c** The flexure of the oceanic crust along the HESC based on summing the individual loads and the progressive flexure. Shaded areas show the flexed oceanic crust (dark blue), the material that infills the flexure immediately beneath the edifice and the load (intermediate blue), and the load (light blue). Red-filled triangles show the depth to the top and base (i.e., Moho) of the flexed oceanic crust based on the seismic data in Fig. 4. Circles show guyots, and triangles show atolls. Green-filled circles based on Hess[85] and black filled circles based on Smoot[86]. Yellow-filled triangles based on Goldberg[87]. TOC = Top of Oceanic Crust. PHA = Pearl and Hermes Atoll. GP = Gardner Pinnacles. FFS = French Frigate Shoals.

indicative[39] of the strength of the elastic portion of the lithosphere and, importantly, the brittle and ductile strength of the entire elastic layer that supports an individual load. Dynamic viscoelastic modeling[24] has demonstrated that such a two-mechanism YSE will recover the 'yield $T_e$' within a 10% error with only modest sensitivity to strain rate.

To estimate the 'yield $T_e$' at the HESC, we first used the 'upstream' flexure in Fig. 3a to calculate the curvature of the flexure due to each individual load cluster. Simple models of loading (Supplementary Fig. S9) show that for a Cartesian coordinate system and volcano loading, there is a maximum positive curvature in the uppermost flexed plate immediately beneath the edifice, which reflects a compressional bending stress, and a maximum negative curvature beneath distal regions of the flanking moats, which reflects an extensional stress. Figure 5a shows the curvatures associated with the 'upstream' flexure. A YSE based on the Byerlee[16] brittle and Goetze[17,18] ductile flow laws based on a coefficient of friction, $\mu_f$, of 0.6, a pre-exponential factor, $A_p$, of $7.0 \times 10^{-14}\,Pa^{-n}\,s^{-1}$, an activation energy, $Q_p$, of 510 kJ mol$^{-1}$, and a cooling plate model[40] was then constructed at 1 km depth intervals from the surface of the plate to the brittle-ductile transition, and the bending moments and curvatures were calculated for different

ages of a plate at the time of loading (Fig. 5b). Other parameters are listed in Supplementary Table S2. We note the strain rate, $\dot{\varepsilon}$, of $10^{-14}\,s^{-1}$ is higher than the $10^{-15}\,s^{-1}$ used by Goetze[17,18]. We justify this value by a consideration of the relaxation that occurs in the Pacific lithosphere during loading and the time it takes for seamount and ocean island formation (Supplementary Table S3). The resulting YSE-derived curvatures were then sampled at the observed curvature and the 'yield $T_e$' were calculated from the bending moment and curvature assuming a Young's modulus of 100 GPa and Poisson's ratio of 0.25 for each profile along the HESC.

The 'yield $T_e$' associated with the compressional bending stress in the uppermost part of the plate beneath each volcanic load is shown in Fig. 5c (black solid line) for Model D and in Supplementary Fig. S10 for Model C. The figures show that as the maximum positive curvature increases, the 'yield $T_e$' decreases, and as the curvature decreases, the 'yield $T_e$' increases, as expected from the YSE. Significantly, the mean differences between the observed $T_e$ and the 'yield $T_e$' are similar for Models D and C ($-16.8 \pm 7.9$ km for Model D and $-17.9 \pm 8.4$ for Model C). Therefore, the observed $T_e$ (colored line) is less than the 'yield $T_e$', suggesting that even when curvature is taken into account and

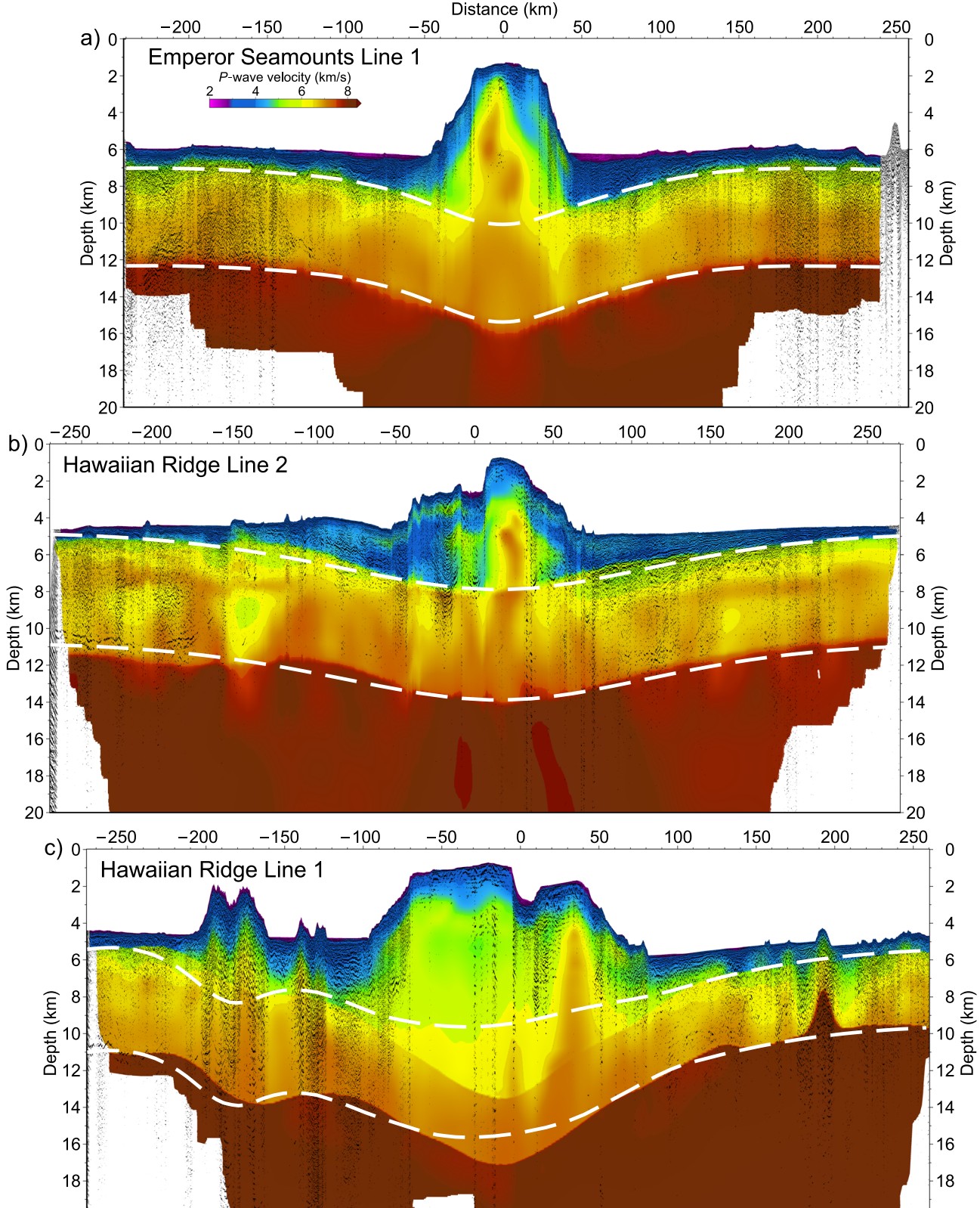

**Fig. 4 | Comparison of the observed *P*-wave seismic velocity and reflectivity structure of the HESC compared to the calculated flexure (thick dashed white lines) on Emperor Seamounts Line 1 and Hawaiian Ridge Lines 1 and 2 (Fig. 1).** The observed *P*-wave velocity is based on Xu et al.[30], MacGregor et al.[31], and Dunn et al.[32]. The reflectivity structure is based on Boston et al.[33]. The calculated flexure is based on the best fit $T_e$ and load and infill density derived from gravity modeling along the profile closest to the 3 seismic transects (Supplementary Table S1). The $T_e$ for Emperor Seamounts Line 1 and Hawaiian Ridge Lines 1 and 2 are 18, 29, and 35 km, respectively. The seismic profiles have been stacked at the center of the volcanic edifice. There is a close agreement between the observed seismic structure and the calculated flexure for Emperor Seamounts Line 1 (**a**) and Hawaiian Ridge Line 2 (**b**). The agreement is not as close for Hawaiian Ridge Line 1 (**c**), even though we have taken into account the contribution to the flexure of both the Cretaceous seamounts and the thinner oceanic crust to the north of Hawaii than to the south[31].

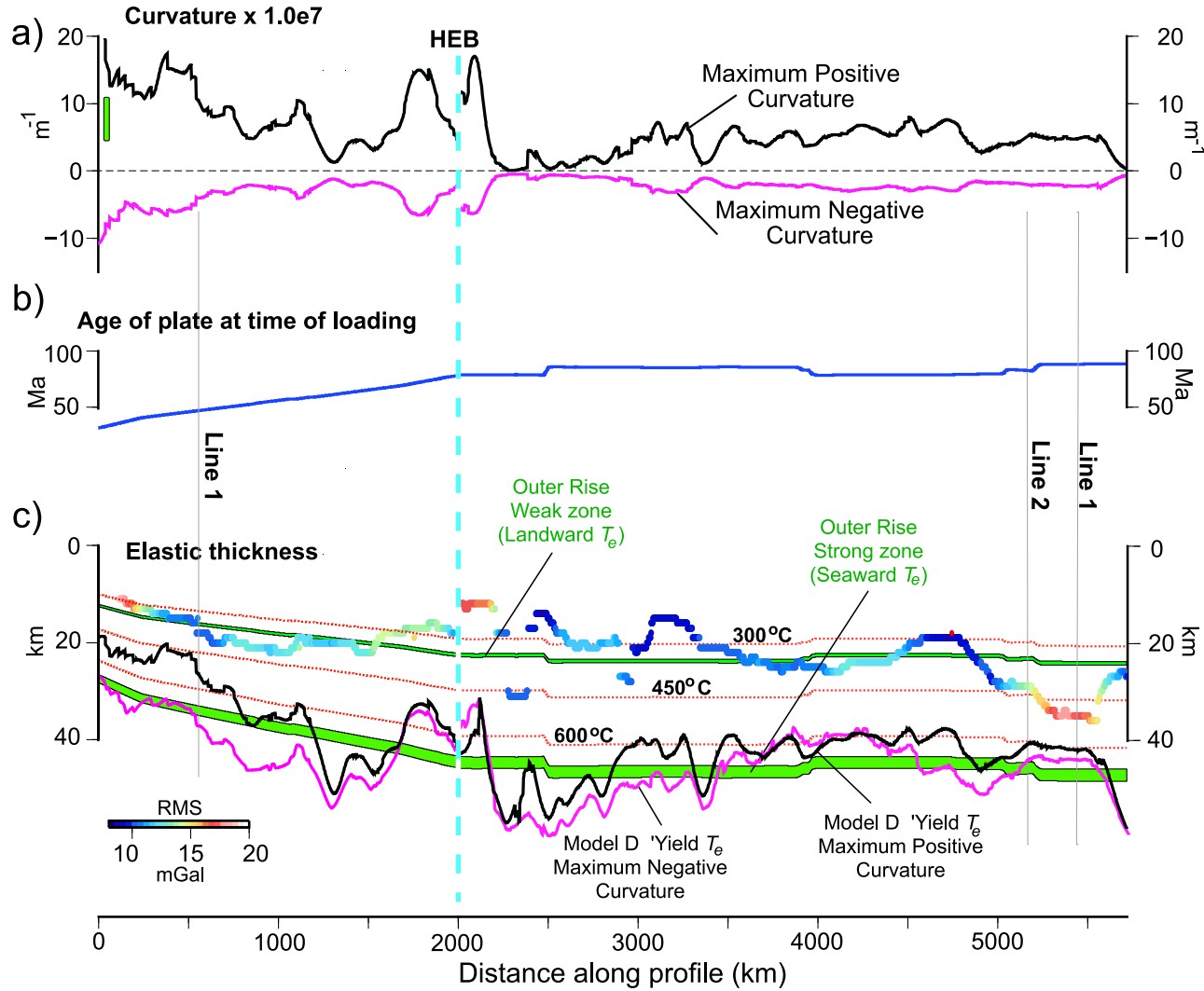

**Fig. 5 | The curvature of flexure and $T_e$ along the HESC based on Model D. a** The curvatures (maximum positive, maximum negative) of flexure due to individual seamount and ocean island loads. The vertical green bar shows, for comparison, the maximum positive curvatures at the CPTOR[21]. **b** Age of the Pacific oceanic plate at the time of loading. **c** Comparison of the observed $T_e$ (colored according to RMS difference between the observed and calculated gravity) to the 'yield $T_e$' (black solid line) derived from a Yield Strength Envelope (YSE) based on the brittle[16] and ductile flow[17,18] laws and the observed maximum positive (black line) and maximum negative curvature (purple line). Thin red dotted lines show the depth to the 300, 450, and 600 °C oceanic isotherms based on a cooling plate model[40]. Green-filled lines show the depth to the controlling isotherms that best fit $T_e$ in the weak zone (340–350 °C, upper curve) and in the strong zone (671–714 °C, lower curve) of the CPTOR[21].

different models are used, the rheology implied by the brittle[16] and Goetze[17,18] ductile flow laws is too strong to explain the observed $T_e$ at the HESC.

Hunter and Watts[21] in their Inversion 'A1' show the average $T_e$ deduced from gravity and bathymetry data at the CPTOR is controlled by the 380–400 °C isotherm based on a cooling plate model[40] and is generally compatible with predictions of the brittle[16] and the Goetze[17,18] ductile flow laws. These results suggest the interior of the Pacific plate may be weaker in its response to long-term loading than its boundaries. However, in their Inversion 'A2' Hunter and Watts[21] explicitly took into account the existence of a weak zone in the pervasively faulted seaward wall of the trench and showed that $T_e$ in the weak zone and the strong zone of the outer rise (oceanward of the weak zone) are given approximately by the depth to the 340–350 °C and 670–710 °C oceanic isotherms respectively. Figure 5c shows that the observed $T_e$ along the HESC follows quite closely the same isotherm that controls $T_e$ in the weak zone in the seaward wall of trenches. The mean and RMS difference between the observed and calculated $T_e$ at the HESC and the weak zone controlling isotherms is reduced to $0.3 \pm 4.6$ km and

$-0.4 \pm 5.0$ km, while the mean difference between the observed and calculated $T_e$ at the HESC and the strong zone controlling isotherm oceanward of the weak zone is increased slightly to $-20.2 \pm 5.0$ km and $-22.5 \pm 5.0$ km.

The results in Fig. 5 suggest some similarity between the $T_e$ recovered along the HESC and the CPTOR[21]. The main difference is in their tectonic setting: the CPTOR weak zone is associated with tension in the uppermost part of the flexed oceanic crust and faulting, while the HESC weak zone is associated with compression in the uppermost part of the flexed oceanic crust and an absence of faulting (e.g., Fig. 4). The observed $T_e$ at the HESC (Fig. 5c, colored line) follows the CPTOR weak zone controlling isotherm while the 'yield $T_e$' at the HESC based on the maximum positive curvature (Fig. 5c, black solid line) generally follows the CPTOR strong zone controlling isotherm. The 'yield $T_e$' in this case is based on the same brittle[16] and Goetze ductile flow laws[17,18], which Hunter and Watts[21] found (in their inversion 'A1'), along with other Low-Temperature Plasticity (LTP) laws such as Mei et al.[20] and Raterron et al.[19], generally account for gravity and bathymetry data at the CPTOR. Interestingly, we found a similar general agreement with

the outer rise strong zone controlling isotherm if we had used the maximum negative curvature (Fig. 5c, purple solid line), which reflects the 'yield' $T_e'$ in the distal parts of the flexural moats that flank volcanic loads along the HESC. Therefore, little difference appears to exist between the strength of the Pacific oceanic plate at its boundary and interior: the plate has a similar long-term strength as it approaches the outer rise seaward of trenches as it has in the flanking regions of large volcanic loads in its interior. It is apparently only in high curvature regions of the seaward wall of a trench and the region immediately beneath a large volcanic edifice where bending stresses are sufficiently high that the flexed oceanic plate weakens, and it is an interesting fact that the plate appears to weaken by a similar amount in both these tectonic settings.

An outstanding question is the origin of these weak zones in the oceanic lithosphere. Watts and Zhong[41] used 2D multilayered viscoelastic models to show that in order to explain the global compilation of oceanic $T_e$ results, it was necessary to reduce the activation energy from 250–400 kJ mol⁻¹ assumed by Karato and Wu[42] to 120 kJ mol⁻¹. Zhong and Watts[23] pointed to other potential sources of weakness in the brittle and ductile flow laws. For example, they found that the 1982 seismic observations of flexure in the vicinity of Oahu could be best explained by the Mei et al.[20] brittle and ductile flow laws by increasing the pre-exponential constant in the LTP laws by a factor of 10⁸ or higher. In contrast, Hunter and Watts[21] showed that Mei et al.[20], as well as other LTP laws[17–19] could explain the free-air gravity anomaly data at the trench-outer rise, confirming that the Pacific plate beneath the Hawaiian Islands was indeed weak when compared to the same age of oceanic lithosphere at the CPTOR. Hunter and Watts[21] suggested the difference between these two tectonic settings might be due to the timescales of loading, with the trenches associated with short-term loads and hence a thicker, stronger plate, and the HESC associated with long-term loads and hence a thinner, weaker plate. Bellas et al.[24,43] used 3D viscoelastic models to confirm previous results[23,41] that the weakness at the Hawaiian Islands could be explained by a decrease in the activation energy from ~500 kJ mol⁻¹ to 320–400 kJ mol⁻¹. Finally, Pleus et al.[35] and Douglas et al.[44] argued that the weakness at the Hawaiian Ridge could be due to a localized decrease in the thermal age of the lithosphere at the time of loading due, for example, to magma assisted flexure.

Figure 6a–d shows that decreases in the frictional coefficient, increases in the pre-exponential constant, decreases in the activation energy and decreases in thermal age either together or in combination could explain the weakness at the HESC. The increases in the pre-exponential and decreases in activation energy required are consistent with previous studies[23,43]. However, there are difficulties with the decrease in the frictional coefficient and the thermal age required. In Byerlee's friction law, the frictional coefficient is 0.60–0.85, and there is support for such values at some strong crustal faults at plate boundaries[45]. In contrast, weakening has been documented at other faults, which result in frictional coefficients as low as 0.1–0.3[46]. Zhong and Watts[23] considered frictional coefficient values of 0.25–0.70 as a source for the weakness, but Fig. 6a shows that despite the absence of faulting and seismicity immediately beneath the volcanic edifice (Fig. 4) much lower values would be required (~0.075) in order to explain the HESC $T_e$ data. Detrick and Crough[47] suggested the Hawaiian Swell was caused by a regional thermal re-setting of the Pacific plate by the Hawaiian plume from 90 to 30 Ma, but the absence of anomalous heat flow[48] and shear-wave velocity anomalies in the lower lithosphere[49] are difficult to explain. Irrespective Pleus et al.[35] and Douglas et al.[44] have suggested a thermal re-setting of the plate that is localized to the 'footprint' of a volcanic edifice by 33–66%. While a similar reset is also required in Fig. 6d, the $T_e$ for Model D increases gradually along the Hawaiian Ridge from ~25 km at Oahu to ~35 km at

Hawaii and weakening of this magnitude is not required to explain the observed seismic and gravity data, even at Hawaii[31], where melts are intruding the crust.

These considerations, together with the lack of evidence of significant amounts of magmatic material at modern trench–outer rise systems, suggest the most likely explanation for the weakening at the HESC, and by inference at the CPTOR, is a load-driven mechanical, not a thermal, one. Figure 6 shows some possible combinations of the brittle and ductile flow law parameters that could account for the HESC $T_e$ data. While the actual mechanism of weakening is not known, we cannot rule out the numerical model results of Gerya et al.[50] who consider weakening at the CPTOR a consequence of brittle normal faulting and ductile damage processes such as those involved in grain-size reduction.

The origin of the weak zone at the HESC, and by inference at the CPTOR, although not clear, occurs within the uppermost part of the Pacific plate that experiences the largest compressional (HESC) and extensional (CPTOR) bending stresses. The weak zone at the CPTOR reflects active loading of the edge of the Pacific plate and must play some role in transferring slab pull forces to the subducting plate[51], while the weak zone at the HESC and its associated downward flexure are effectively 'frozen in' following volcanic loading. The Pacific plate is known to retain a memory of past geological events[52] so the weak zone at the HESC could be considered a permanent feature of the plate, which, like a transform fault and fracture zone[53,54] or a detachment fault[55], may act as a site for subduction initiation. Indeed, Lallemand and Arcay[56] have proposed that seamount chains or oceanic plateaus are sites of weakness that may control where subduction initiates, citing the 400-km-long Afanasy Nikitin seamount chain in the Central Indian Ocean as an example where there is evidence for both low $T_e$[57] and intraplate deformation[58]. Once initiated, a suprasubduction zone may develop where an oceanic plate is subducted beneath another plate with an actively spreading ridge and maybe a trench, forearc, and volcanic arc. Seamounts and ocean islands and their associated downward flexures may be carried over an outer rise on a subducting plate towards the trench, decapitated, fragmented, and, depending on their tectonic setting (i.e., whether they formed on-ridge or off-ridge), be either carried down with the subducting plate or accreted to a forearc[59]. When spreading ceases or a rifted continental margin comes into contact with a forearc region, the fragments may become part of an ophiolite sequence. Indeed, former seamounts and ocean islands have been described from a number of collisional orogens within ophiolite sequences in Oman[60], China[61], Turkey[62], Iran[63], and USA[64] some of which have carbonate caps and a substrate with a geochemical affinity to seamount and ocean island basalts and may be associated with mantle plumes.

In conclusion, gravity and seismic modeling suggest that $T_e$ increases gradually along the HESC from low values at the Emperor Seamounts to high values at the Hawaiian Ridge and is controlled approximately by the depth to the same oceanic isotherms that describe the weak zone at deep-sea trenches (340–350 °C). Calculations based on the curvature of flexure and the Byerlee brittle and Goetze ductile flow laws suggest that the HESC 'yield $T_e$' is controlled by the same isotherms that describe the strong zone at the outer rise seaward of trenches (670–710 °C). These observations suggest more similarities than differences between the long-term rheological properties of the Pacific oceanic plate in its interior and its boundaries. For example, the plate that flanks the HESC weak zone is strong, as is the plate approaching the CPTOR. The fact that weak zones can pre-exist at hotspot-generated seamount chains within the interiors of large, otherwise rigid, plates may, we speculate, facilitate intraoceanic subduction in response, for example, to changes in plate motion and eventually lead to their break-up.

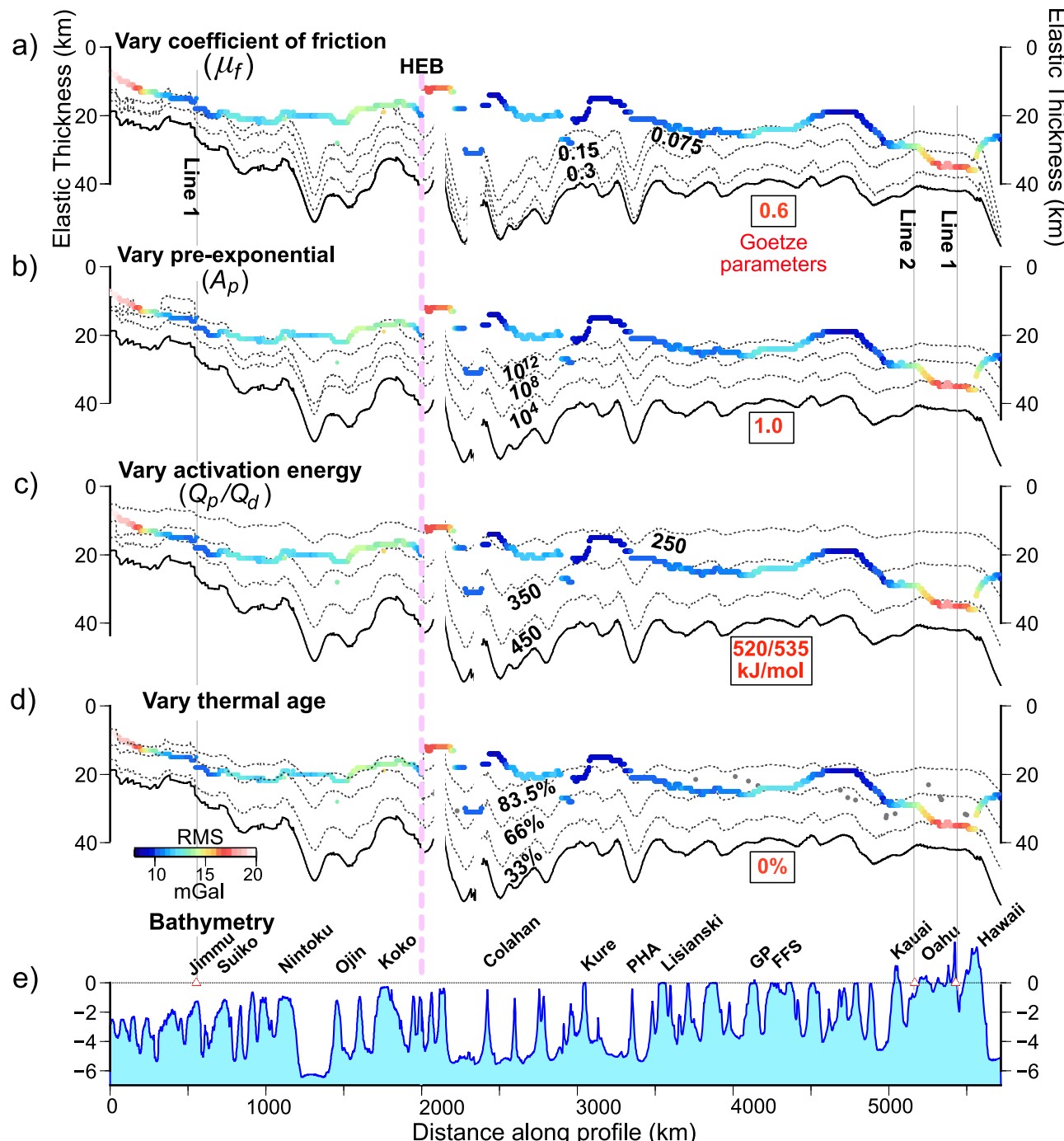

**Fig. 6 | Comparison of the observed and calculated $T_e$ based on the YSE.** The observed $T_e$ has been derived from 3000 bathymetry and gravity profiles and the calculated $T_e$ has been derived from the brittle[16] and ductile flow[17,18] laws based on a uniform strain rate of $10^{-14} s^{-1}$, a thermal age at the time of loading, and a curvature of flexure as shown in 5a, b along the HESC. Dotted black lines show the calculated 'yield $T_e$' based on different brittle and ductile flow parameters. Numbers in bold red font show the original parameters as used in Goetze[17,18] (Supplementary Table S2). Numbers in bold black font identify variations in these parameters. **a** Calculated $T_e$ based on reducing the coefficient of friction, $\mu_f$, from 0.6[17,18] to 0.3, 0.15 and 0.075. **b** Calculated $T_e$ based on increasing the value of the pre-exponential constant, $A_p$, by factors of 1, $10^4$, $10^8$ and $10^{12}$. The factor of 1 corresponds to the original parameters as used in Goetze[17,18]. **c** Calculated $T_e$ based on reducing the activation energies, $Q_p$ and $Q_d$, from 520 and 535 kJ mol[17,18] to 450, 350 and 250 kJ mol$^{-1}$. **d** Calculated $T_e$ based on reducing the thermal age of the plate at the time of loading by factors of 0%[17,18] to 33%, 66% and 83.5%. **e)** Bathymetry along the axis of the HESC derived from the SRTM15 V2.4 15 × 15 arc s bathymetric grid[65].

## Methods

Modeling was carried out using the Fast Fourier Transform (FFT) for the flexure and a combined analytical and FFT method for the gravity anomaly. The first step was to isolate the volcanic loads that drive the flexure. This was achieved by selecting a 1200 × 1200 km window centered on the chain and applying an Optimal Robust Separator (ORS)[37] filter to a SRTM15 + V2.4 bathymetry grid[65] and the V29.1 satellite-derived free-air gravity anomaly grid[66]. We found that a filter width of 346 km for the Emperor Seamounts and 450 km for the Hawaiian Ridge most satisfactorily isolated the regional bathymetry

and gravity while at the same time maximized the amplitude of the load and the gravity anomaly associated with flexure. The correlation between the regional bathymetry and gravity anomaly at the young end of the HESC (Supplementary Fig. S6) is close such that the gravity to bathymetry ratio is ~21–27 mGal/km, similar to that derived by Watts[67] and used by McKenzie[68], suggesting that ORS filtering has removed much of the effect of the Hawaiian swell and its compensation. The difference in filter widths between the two chain segments was attributed by Watts et al.[22] to the shorter wavelengths of flexural isostasy associated with the Emperor Seamounts than the Hawaiian Ridge.

Once isolated, the loads were used to calculate the flexure and gravity anomaly. Since sample ages[69] and oceanic crustal ages based on magnetic anomaly identifications[70] suggest the age of the lithosphere at the time of volcano loading varies from ~32 Ma at Detroit Seamount to ~78 Ma at the Hawaiian-Emperor Bend (HEB) to ~90 Ma at Hawaii, it is likely that $T_e$ varies spatially along the HESC. The Fast Fourier Transform (FFT) method of computing gravity[71] and flexure[52] usually assumes uniform $T_e$ and so is strictly not applicable to such situations. We circumvent this limitation by computing 2D solutions based on the FFT for the top and bottom of the flexed oceanic crust for a wide range of elastic thickness, $T_e$ values (from 0 [i.e., Airy isostasy] to 50 km in steps of 1 km) and determining the best-fitting $T_e$ on 3000 closely spaced (2 km) profiles (Fig. 1, inset) using only data within a 1200 × 1200 km window centered on the chain (as was carried in the ORS analysis). This approach enabled $T_e$ to be a free parameter that could be determined as a function of position along the chain.

The gravity anomaly, as previous studies have shown, is sensitive to the flexure as well as to the bathymetry and the densities assumed for the load, the root infill (defined here as the material that infills the flexure immediately beneath the volcanic load), and the flanking moat infill. We construct here a 3D bathymetry-dependent model in which the load density varies in depth within the volcanic edifice, similar to the model used by Wessel et al.[72] at Jasper Seamount and Watts et al.[73] at Jimmu Guyot. In both these examples, the density structure was constrained by the seismic P-wave velocity derived from active source seismic experiments[30,74] using empirical relationships between P-wave velocity and density as summarized in Brocher[75]. The lowest density on the load flank and the highest density in the load core were specified on the basis of the P-wave velocity structure to be 2100 and 3020 kg m$^{-3}$ respectively.

A number of the seamounts along the HESC are guyots and so have been subject to significant amounts of mass wasting, including the loss of their summits through wave action. We use a 'load boost' to allow consideration of the mass missing (Supplementary Fig. S11) in the present-day bathymetry that may have contributed to the gravity anomaly and flexure at the time of volcano loading. We present two models here: one in which the load density varies spatially with the bathymetry and there is a 'load boost', γ, of 1.35 (i.e., Model D[73]) and one in which the load density varies spatially and γ = 1.00 and so there is no 'load boost' (i.e., Model C[73])."

The remaining densities concern the material that infills the flexure in the flexural depressions flanking the seamounts and ocean islands (the moat infill) and the material that infills the flexure immediately beneath the load (the root infill). Since these densities may like $T_e$ also vary along the HESC and are again not well determined by the limited seismic data we therefore also searched for them along each of the profiles.

The gravity anomaly calculation is the sum of contributions from the bathymetry, the load, infill and the flexed oceanic crust[73]. Of these contributions the load calculation was the most computer intensive since it involved calculating the 3D gravity effect of a series of stacked rectangular prisms[76,77] (solid light blue line in Supplementary Fig. 12a).

We performed this calculation using GMT's `gravprisms`[78] assuming that each stack of prisms extended down from the seafloor to a reference depth corresponding to the base of the bathymetric load of 6 km and 5 km for the Emperor Seamounts and Hawaiian Ridge respectively. We note that the gravity effect of the load only needed to be computed once for the reference density variation, since we can simply multiply the resulting gravity anomaly by the Parker[71] algorithm for other profiles, as the FFT of the gravity anomaly is a linear function of the density contrasts.

The final step is to compute the gravitational effect of the infill and the flexed oceanic crust. We will use 2D FFT-based solutions[71] as implemented in GMT's `gravfft` module. We used 2D because the surfaces of flexure are deep, and we found that the differences in the gravity anomaly between the 3000 profiles to be small. Here, in order to accurately include the flexural depression and the bulge, we create 7 different surfaces at different distances below the seafloor, each of which has a constant density contrast across it (Supplementary Fig. S12). Since we aim to be as exact as possible and because the FFT method is most stable when the density contrast is constant, we treat the different density contrasts related to the flexural compensation as separate interfaces.

## Data availability

The minimum and maximum curvatures of flexure and the 'best fit' elastic thickness, $T_e$ for Model D, plotted in Figs. 3, 5, 6 are available from https://doi.org/10.6084/m9.figshare.29446586. The seismic refraction and reflection data plotted in Fig. 4 are available from https://www.marine-geo.org/tools/entry/MGL1806 (Lines 1 and 2, Hawaii) and https://www.marine-geo.org/tools/entry/MGL1902 (Line 1 Emperor).

## Code availability

The illustrations and most of the calculations in this paper were produced by the Generic Mapping Tools version 6.5[78]. Source code and installers for GMT are available separately at www.generic-mapping-tools.org under the GNU LGPL license 3 or later. The scripts used to carry out the ORS analysis and to determine the best fit $T_e$ structure are available from https://doi.org/10.6084/m9.figshare.29446775.

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

## Acknowledgements

The acquisition of the seismic data that underpins this research was funded by the US National Science Foundation research grants OCE-1737245 to DJS and ABW and OCE-1737243 to RD. ABW was supported by a Leverhulme Trust Emeritus Professorship and CX by the Ocean University of China. We are grateful to the captain, crew, and science party of the R/V *Marcus G. Langseth* during legs MGL1806 and MGL1902 and to the ocean bottom seismic groups from Woods Hole Oceanographic Institution, Scripps Institute of Oceanography, and GEOMAR Helmholtz Centre for Ocean Research who collected the seismic data used to construct Fig. 4.

## Author contributions

A.B.W. and D.J.S. conceived the idea. D.J.S., B.B., R.D., and A.B.W. participated in data acquisition and reduction. P.W. created software. A.B.W. and C.X. constructed the models. A.B.W. wrote the original paper. A.B.W. and C.X. made the figures. All authors discussed the results and implications.

## Competing interests

The authors declare no competing interests
