## [Transparent Peer Review file · Nature Communications]

New seismic and gravity constraints on plate flexure and mantle rheology along the whole Hawaiian-Emperor seamount chain

Corresponding Author: Professor Anthony Watts

Version 0:

Reviewer comments:

Reviewer #1

(Remarks to the Author)

Review of "New seismic and gravity constraints on plate flexure and mantle rheology along the whole Hawaiian-Emperor seamount chain".

In this manuscript, the authors investigated the elastic thickness of the Pacific Plate along the Hawaiian-Emperor seamount chain, using seismic and gravity observations as constraints. They evaluated the elastic thickness along 3000 profiles with a spatial interval of 2 km, and found that the elastic thickness along the seamount chain is generally consistent with those inferred for the weak zones at the seaward wall of circum-Pacific trenches, lower than those inferred for the interior of the Pacific Plate and the strong zones of the circum-Pacific trench outer rise. This finding is of interest to the broad audience of the Earth Science, as it helps constrain the plate rheology. It also provides a reliable estimation on the volume flux of the Hawaiian plume since the Cretaceous, which should provide some insights on the evolution of the Hawaiian plume.

I support the publication of this manuscript after some minor revision. The manuscript is very well written with the figures very well illustrated. I enjoyed reading this manuscript. However, I must disclose that I am not an expert in plate flexure modeling. Therefore, I am not able to comment on the technical details, for example, whether it is a good approximation to use 2D elastic plate modeling despite the 3D nature of the islands on the Hawaiian-Emperor seamount chain, and whether the misfit between the modeled flexure and the seismically constrained flexure at Hawaiian Ridge Line 1 should be considered "acceptable". It would be important to make sure that the manuscript is also reviewed by someone with such a background. I believe some of the novelty lies in the technical aspects and the thorough investigation using variable model parameters. Below I list some places that confuse me, which I hope can be helpful to make the manuscript more accessible to nonspecialists.

1. When computing the loading, the density variations within the mantle lithosphere and below is not considered. There should be some thermal erosion caused by the plume. Additionally, at the Hawaiian Islands, the positive buoyancy of the mantle plume may also partially lift the lithosphere. I am wondering whether these density variations would affect the estimation of the elastic thickness, and how large the impact would be.

2. The inference on plate rheology to me is the most important implication of this study. The authors have provided multiple alternative explanations, although most of them seem not satisfying to the authors. I am thinking if the distinct elastic thickness is related to the time scale of the loading processes. Is it possible to provide and compare the strain rates of the lithosphere when the plate bends at the trench versus when the plate flexures during magma loading at the HESC. Recent studies have also invoked grain size reduction during slab bending at the trench to account for the weakening of slab. I am wondering if grain size reduction could be a candidate for the thin elastic thickness at HESC.

3. It would be nice to give a clear definition of the yield T_e . I realize that it is calculated from the yield strength envelop, but I am not sure how this envelop is converted to a thickness, and which part of the envelop is considered mechanically strong.

Minor issue: In Fig. 1, "Blue lines show the 83 and 121 Ma isochrons" seem to be missing.

Reviewer #2

(Remarks to the Author)

This study investigates the strength of the Pacific plate along the trail of seamounts associated with the Hawaiian hotspot. The authors make use of satellite derived gravity, bathymetry, and seismic experiments to develop new estimates of the strength of the Pacific plate along the chain of seamounts, taking into account many confounding factors in a thorough and detailed manner. The paper is well-written throughout, and the supporting animations in particular are valuable.

My main two concerns relate to points made in the summary, (1) that the new results contradict previous studies and (2) that the new results have significant implications for understanding subduction initiation. Detailed comments on both issues are provided below. My overall assessment is that the manuscript may be suitable for publication in Nature Communications if the authors can clarify these points and confirm the novelty and significance of the work.

Regarding two of the points emphasized in the Summary:

“The new results show that, contrary to previous predictions, T_e changes gradually along the chain....”

This phrase sets the expectation that the new results are revealing a flaw in current thinking, yet it is not clear to me from within the manuscript whether this is really the case. On line 52 we are told that many previous studies yield an abrupt change in T_e at the HESC bend. The most recent of these studies is 1993. Many more recent studies (e.g. Kalnins and Watts, 2009; Lu et al, 2021) don't show an abrupt change at the bend. Further, subsequent text in the 'Results' section states a broad agreement between the new results and some previous studies, so that in the end it is not clear how the new 2D profile model results are providing novel insights. I suggest the authors aim to clarify this point.

“...weak zones can exist within the interior of a large, otherwise rigid, plate and may, we speculate, facilitate the initiation of intra-oceanic subduction in response, for example, to changes in plate motion and eventually lead to their break-up.”

This is an interesting idea, and I had the expectation that the main text would develop it further. However, only the paragraph beginning on line 465 touches on it again. There is a huge literature on subduction initiation that goes completely ignored, and at the least it would be useful for the authors to articulate how they imagine the interpreted weak zones to contribute to existing paradigms - either in a general sense, or making suggestions for specific examples in the recent past (see Gurnis et al, 2004; Lallemand and Arcay, 2021, and many others) where it is at least plausible that weak zones associated with hotspot trails were an important factor.

A related question would be, does the new proposal make testable predictions that would be valuable to other communities. For example, there are many workers studying subduction initiation by mapping and sampling ophiolites, is the implication of this study that we might expect to see certain things in ophiolites interpreted to record some aspects of subduction initiation that is characteristic of weak zones associated with hotspot trails?

Further related question - what can we say about the nature of weak zones related to volcanic trails in general, as opposed to just the HESC specifically? For example, it should be possible to produce T_e estimates for Louisville and other trails (or use existing ones), considering these could reveal something about whether such weak zones are likely to be common through Earth history or limited to only a subset of trails with Hawaii-like characteristics.

Regarding writing style - I would suggest large sections of text need to be made more accessible for a Nature Communications article, given the target audience should be wider than (for example) GJI or JGR. Certain concepts such as the 'upstream' and 'downstream' flexure are introduced with little context and would require most readers to immediately go and read other sources for background for the text here to be understandable.

Regarding uncertainty propagation - I think the authors have done a decent job of considering the many various choices and assumptions that must be made to arrive at T_e estimates in this way. Nonetheless, the final presented values generally appear as a single line of best-fitting T_e versus distance (e.g. Figure 3). We are therefore left to wonder how the magnitude of the variations in this series compares to the uncertainty associated with any or all of the choices at each step. For example, I could imagine that fitting for each profile could yield a range of possible fits that give some kind of confidence region, which together would give an envelope around the best-fitting curve. Of course this would also depend on some assumptions but I think it would be useful to illustrate the magnitude of such envelopes.

Other technical comments:

Missing some context, given that we need to understand what constitutes a 'weak zone' in the context of surrounding 'normal' oceanic lithosphere.

I am surprised there is no mention of the work of Lu et al (2021), "What Controls Effective Elastic Thickness of the Lithosphere in the Pacific Ocean", JGR. Full disclosure, I have no particular skin in the game in terms of contrasting interpretations, rather I just want to understand the origins of any contrasting ideas in the current literature. My impression is that the current submission brushes some issues under the carpet. Low T_e at subduction zones seems to be a debated issue, yet the text glosses over this. The broader issue to me is that (in furtherance of comments made above) the results presented in the current submission are very much focussed on the HESC without much context (at least in terms of the figures). Consider for example in figure S9, we see plotted some estimated volume fluxes for the HESC with the Wessel

(2016) estimates plotted for comparison. This is useful context (even though the flux is a relatively minor issue here. What I would like to see is a much better attempt to illustrate current ideas on T_e within Pacific plate lithosphere (or better, oceanic lithosphere in general) with the new results for the HESC. While all the results appear to be reasonable in themselves, there is a large conceptual leap to the 'significance for subduction initiation' which needs to be bridged.

Final point, I wonder whether another factor not mentioned relates to the other hotspot trails on the Pacific plate which transect the HESC. For example, older trails transect the HESC (Shatsky, Hess, Musicians), we should expect these to also influence T_e ?

Reviewer #3

(Remarks to the Author)
Review

This study investigates the lithospheric flexure and rheological properties of the Pacific plate along the Hawaiian-Emperor Seamount Chain (HESC) using gravity and seismic data constraints. The authors analyzed 3000 profiles spaced 2 km apart and applied 2D/3D forward modeling to estimate effective elastic thickness T_e . Contrary to previous interpretations suggesting an abrupt change in T_e at the Hawaiian-Emperor Bend (HEB) or kink, the study finds a gradual variation, with low T_e at the Emperor Seamounts and high T_e at the Hawaiian Ridge. The modeling incorporates recent seismic constraints and considers varying densities of volcanic loads and infill materials.

Results indicate that T_e correlates better with the 340–350°C oceanic isotherm, suggesting a mechanical, rather than thermal, weakening of the lithosphere due to volcanic loading. Comparisons with yield strength envelopes (YSE) based on brittle and ductile rheology show that traditional flow laws are too strong to explain the low observed T_e . The findings imply that weak zones may exist within tectonic plate interiors, potentially preconditioning them for subduction initiation and plate break-up, offering insights into intraplate deformation and mantle dynamics.

After reading the manuscript, I found several aspects that are either insufficiently explained or entirely omitted (specially quantification of parameters through the text), which the authors should address to better convince the reader. Most of these issues are minor, while others are of moderate. Below, I outline several key points:

Abstract

..... "While low T_e at the Emperor Seamounts and high T_e at the Hawaiian Ridge are expected because of their differences in volcano and plate age,"

How are "low" and "high" T_e defined in this context? Please provide quantitative values or ranges—otherwise, the terms remain subjective and open to interpretation.

..." T_e changes gradually along the chain"

It is stated that T_e changes gradually from one value to another, but the specific values are not provided. Could you please clarify what X and Y are? Without this information, the statement remains vague.

Introduction and motivation

Line 44. "thin elastic or viscoelastic plate with uniform"

What exactly is meant by "thin" in this context? While elasticity theory assumes that T_e is much smaller than the plate length, previous studies have reported specific ranges of T_e values for the Hawaiian Ridge and Emperor Seamounts. This information should be clearly explained and acknowledged in the Introduction to provide proper context.

Line 49. "Cretaceous Normal Polarity oceanic crust"

This explanation is clear and appropriate for geoscientists. However, since Nature Communications reaches a broader scientific audience, please consider adding the age range in parentheses—for example, "(121–83 Myr old)"—to provide clearer context for readers from other disciplines.

Lines 52-54: "While subsequent studies 5-10 have generally confirmed these results, they reveal that T_e changes abruptly at the HESC bend which is not expected from the volcano load and plate ages"

The bend appears more like a kink—an abrupt change in the Pacific Plate's azimuth. When did this kink occur? According to Wessel & Kroenke (2008, JGR), it happened between 53.4 and 47.9 Ma. This fundamental detail should be clearly stated in the Introduction.

Figure 1.

- Some isochron values on the oceanic plate are missing. At a minimum, please include isochrons at 50 Myr intervals for

better reference.

- Additionally, key ages of the Hawaiian Ridge and Emperor Seamounts should be labeled. There are well-established geochemical and geochronological data available for several locations along the HE and ES chains that would help readers understand the age difference between the Pacific Plate and the seamounts. This would effectively illustrate the classical concept of “the age of the plate at the time of volcanic emplacement or loading,” which is critical for interpreting flexure.

Lines 122-123. “for example, with the Hawaiian swell while at the same time maximised the amplitude of the load (Supplementary Figure S2).”

I'm not sure the Hawaiian swell signature can be dismissed a priori. McNutt & Bonneville (2000, G-Cubed) and Watts (Chapter 5, Flexure and Isostasy of the Lithosphere, 2nd ed.) highlight the significance of both buried and subsurface loads associated with underplating of anomalously thick crust. This anomalously thick crust is less dense than the surrounding mantle and generates buoyant stresses that contribute to swell formation.

While the volcanic load—with its higher density contrast relative to seawater—likely plays a dominant role in downward flexure, the contribution of the buried load should not be overlooked. Its effect should be quantified, at least along profiles constrained by seismic models that provide density distributions and Moho geometry.

Although I expect the results may not vary significantly due to the relatively high effective elastic thickness (T_e) of approximately 20 km along the Hawaiian-Emperor Seamount Chain (HESC), this assumption needs to be tested explicitly along selected profiles and shown in Methods of Supporting Information.

Figure 3c.

At first glance, this figure appears somewhat unrealistic. Typically, we would expect infill material to accumulate along the flanks of the oceanic ridge, adjacent to the seamount. However, Figure 4 shows that the volcanic edifice extends continuously from the flexed oceanic crust all the way to the core of the seamount. An exception is seen along the Vp model along Line 2, where infill material occupies approximately half of the elevated topography.

This raises the question: how different is the density of the volcanic edifice compared to that of the infill material? If the densities are similar, then Figure 3c could serve as a useful—though simplified—proxy for the system.

That said, it is important to recognize the limitations of the model. These include assumptions about uniform density, simplified geometry, and the omission of sediment compaction or lateral variations in lithology. Such simplifications may lead to discrepancies between the model output and the actual structure, particularly in areas where infill material dominates or where crustal flexure varies significantly.

Please explain the limitations and uncertainties of the model. It appears that a constant density is assumed for each material body, which may not reflect natural variability.

Figure 4. Please clarify: were the dotted lines calculated using a constant or variable effective elastic thickness (T_e)? What range of T_e values best fits the data in panels (a), (b), and (c)? This information should be clearly specified in the figure caption.

Figures 5 and 6. It is surprising that the “yield T_e ” is approximately twice as large as in the purely elastic model, even though $T_e(x)$ is allowed to vary and decrease in regions of high plate curvature in the elastic model. The yield T_e —or more precisely, the yield strength envelope (YSE)—is known to depend on plate curvature, as shown in Figure 5, but it is also highly sensitive to several parameters: the strain rate (s), activation energy (Q), the stress exponent in the ductile flow law (n), constant of the material (A), the friction coefficient (p), the fluid pore pressure ratio (A), and the thermal structure ($T(z)$). An explicit expression for the yield deviatoric stress should be included in the Supporting Information or Methods section. There are various formulations beyond the exponential laws, and the specific choice is critical for deriving a higher “yield T_e ” compared to the purely elastic model.

Later, the authors mention that “...the brittle and ductile flow laws are too strong to explain the observed T_e at the HESC...”. At this point in the manuscript, they should explicitly state the values of the critical parameters used (s , A , Q , p , n , and A), rather than waiting until Figure 6 to discuss the calibration of p , n , Q , and $T(z)$ to match the observations.

A few questions arise from this analysis:

- Why was the thermal age modified? It is generally well constrained, except in specific regions affected by plume-related thermal rejuvenation.
- What is the impact of decreasing s from 10^{-14} to 10^{-15} or 10^{-16} s^{-1} ? In fact, Hunter & Watts (2016) used $s = 10^{-16} \text{ s}^{-1}$. What strain rate was used in Figure 5c for the “strong” Pacific Plate?
- What value of the fluid pore pressure ratio (A) was applied in the brittle portion of the YSE calculations?
- In Figure 6 shows the variation of Q are shown, but it is unclear what value of A was used in the calculations.

The geological cause of the apparent plate weakening remains unclear. From the perspective of the YSE, is this weakening primarily attributed to changes in A and n , or to a lowers? Which factor is more likely responsible?

Finally, it would be very helpful to include a brief sentence indicating the typical range of plate curvature values in outer-rise settings, and how these compare to those associated with flexure due to seamount loading.

The last sentence "... we speculate, facilitate intra-oceanic subduction in response, for example, to changes in plate motion and eventually lead to their break-up" is identical to the last sentence of the Summary/Abstract.

Although this is a speculation, plate breakup is typically accompanied by thermal upwellings and extensional stresses. Could you elaborate a bit more on this speculation and clarify the underlying mechanisms you envision?

Comment: I agree with the authors that the weakening observed at both the HESC and CPTOR is primarily a consequence of mechanical loading rather than thermal processes. This suggests a role for spatial variations in the parameters governing both brittle and ductile deformation, as has been proposed.

Videos.

The videos are very good and illustrative. However, the physical units on both the horizontal and vertical axes are missing.

Version 1:

Reviewer comments:

Reviewer #1

(Remarks to the Author)

The authors have well responded to my comments.

I think the manuscript is more accessible to the general audience now.

I have no further comments.

Reviewer #2

(Remarks to the Author)

I have not much to add based on the revised manuscript. The authors have made a good attempt to respond to comments, just a few new issues:

Figure 5 and related text, paragraph 417:

I would find it useful to make a more explicit connection between the text and the two alternative green curves, provide a basic explanation for what the 'weak zone' and 'strong zone' curves are and why they are significant here.

Line 437 - consider shortening sentence.

Line 511 - consider shortening sentence.

Line 529: "...seamount chains or oceanic plateaus may initiate subduction..."

Consider the wording here, I suspect you mean to convey that these features are sites of weakness that may control where subduction initiates (driven by forcing from elsewhere). Rather than being the driver themselves.

Reviewer #3

(Remarks to the Author)

The authors have addressed my suggestions, and the manuscript is now much improved. I have no further requests.

Eduardo Contreras-Reyes

21 July 2025

In the following, black font shows the comment/query of the Reviewers and Editor and the red font our response.

Reviewer 1

I support the publication of this manuscript after some minor revision. The manuscript is very well written with the figures very well illustrated. I enjoyed reading this manuscript.

We are pleased the reviewer supports the publication of our paper and found our paper “very well written....and illustrated”.

However, I must disclose that I am not an expert in plate flexure modeling. Therefore, I am not able to comment on the technical details, for example, whether it is a good approximation to use 2D elastic plate modeling despite the 3D nature of the islands on the Hawaiian-Emperor seamount chain,

We used both 2D and 3D modeling techniques. 2D modeling was used to calculate the gravity anomaly of the flexed top and base (i.e., Moho) of the oceanic crust (the main contributor to the negative part of the gravity anomaly) along each profile. This was because the surfaces of flexure are deep (> 6 km), and we found little or no change in either the gravity anomaly or flexure between the 3000 closely spaced (2 km) profiles. 3D modeling was used, however, to calculate the gravity effect of the load (the main contributor to the positive part of the gravity anomaly), the flexures due to individual ocean island and seamount load clusters, the progressive flexure, and the curvatures of flexure.

We have made our choice of 2D and 3D techniques clearer in the revised text at the end of the Gravity and Flexure Modeling section.

and whether the misfit between the modeled flexure and the seismically constrained flexure at Hawaiian Ridge Line 1 should be considered “acceptable”.

We consider the fit between the modeled flexure and the seismic data along Line 1 is “acceptable” given that this is the young end of the Hawaiian-Emperor Seamount Chain (HESC), and it is possible that isostatic equilibrium here is still not complete. For example, Line 1

Professor Tony Watts FRS - Professor of Marine Geology and Geophysics

Tel: +44(0)1865 272032

Fax: +44(0)1865 272072

Email: tony@earth.ox.ac.uk

<http://www.earth.ox.ac.uk/~tony/watts>

intersects the ‘donut’ of earthquakes centered on the big island of Hawaii (Klein, 2016), which indicates active deformation and, possibly, a broken rather than a continuous elastic plate as we modeled.

We have pointed this out and added the Klein reference in the revised caption to Figure 4.

When computing the loading, the density variations within the mantle lithosphere and below is not considered. There should be some thermal erosion caused by the plume. Additionally, at the Hawaiian Islands, the positive buoyancy of the mantle plume may also partially lift the lithosphere. I am wondering whether these density variations would affect the estimation of the elastic thickness, and how large the impact would be.

The reviewer is correct in pointing out density variations in the mantle due, for example, to a mantle plume and its impact on the lithosphere were not directly considered in our study. However, Optimal Robust Separator (ORS) filters were applied to both the observed bathymetry and free-air gravity anomaly data in order to remove the relatively long-wavelength effect of a mantle plume and its impact on the lithosphere and isolate the relatively short-wavelength bathymetry and gravity anomaly associated with flexure.

To demonstrate this we added the ORS filtered free-air gravity anomaly to Supplementary Figure S2. The figure reveals a strong correlation between the ORS filtered bathymetry and gravity anomaly in the region of the Hawaiian swell. The maximum ORS filtered bathymetry and gravity anomaly on the swell crest is 0.99 km and 20.58 mGal, respectively which implies a gravity to bathymetry ratio of ~ 21 mGal km⁻¹, similar to the value estimated by Watts (1976) of 22 mGal km⁻¹ and attributed by him to some form of convective upwelling in the mantle that supports the swell.

We believe therefore that ORS filtering satisfactorily isolates the bathymetry and gravity anomaly associated with the mantle plume that is presently located beneath the youngest volcano in the HESC and so after subtraction from the observations will have little or no effect on our recovery of T_e .

The inference on plate rheology to me is the most important implication of this study. The authors have provided multiple alternative explanations, although most of them seem not satisfying to the authors. I am thinking if the distinct elastic thickness is related to the time scale of the loading processes.

We tested a number of possible causes of the weakening in the brittle and ductile fields, including reduction of the coefficient of friction, increasing the pre-exponential constant, decreasing the activation energy, and decreasing the thermal age of the lithosphere. Of these we found that the values of the coefficient of friction and thermal age required to explain the T_e are unreasonably low and high respectively, although some combination of these parameters maybe possible.

We believe that the cause of the weakening is mechanical, not thermal, and that it is most likely the result of an increase in the pre-exponential constant, a decrease in the activation energy or some combination of these parameters.

While the loading time scale may be a consideration at trenches where deformation is active, it is not considered significant along the HESC, except perhaps at the youngest end of the chain. This is because the spread of sample ages suggests individual seamounts and ocean islands take at least a few Myr to form and by this time relaxation is essentially complete.

Is it possible to provide and compare the strain rates of the lithosphere when the plate bends at the trench versus when the plate flexures during magma loading at the HESC. Recent studies have also invoked grain size reduction during slab bending at the trench to account for the weakening of slab. I am wondering if grain size reduction could be a candidate for the thin elastic thickness at HESC.

It is possible to estimate the strain rate at a trench from the distance to the neutral surface in the flexed plate, the change in the curvature of flexure with distance, and the subduction velocity. For example, Hunter (2015, PhD thesis) and Hunter & Watts (2016) derived strain rates of $10^{-15.1}$ to $10^{-16.3}$ s⁻¹ for circum-Pacific trenches. We believe that grain size reduction in the ductile layer (e.g. as described for example by Gerya et al. 2021 at trench – outer rises) could indeed be a candidate for the weakness at the HESC and so we have added a reference to this process in the revised text.

It would be nice to give a clear definition of the yield T_e . I realize that it is calculated from the yield strength envelop(e), but I am not sure how this envelop(e) is converted to a thickness, and which part of the envelop(e) is considered mechanically strong.

We have better defined the yield T_e where it is first mentioned in the Discussion section and added a reference to Watts & Burov (2003), as suggested by the Reviewer.

Minor issue: In Fig. 1, “Blue lines show the 83 and 121 Ma isochrons” seem to be missing.

We have added the missing blue lines, as suggested by the Reviewer.

Reviewer 2

This study investigates the strength of the Pacific plate along the trail of seamounts associated with the Hawaiian hotspot. The authors make use of satellite derived gravity, bathymetry, and seismic experiments to develop new estimates of the strength of the Pacific plate along the chain of seamounts, taking into account many confounding factors in a thorough and detailed manner. The paper is well-written throughout, and the supporting animations in particular are valuable.

We are pleased the Reviewer considers the paper “well-written throughout” and the supporting animations “valuable”.

My main two concerns relate to points made in the summary, (1) that the new results contradict previous studies and (2) that the new results have significant implications for understanding subduction initiation. Detailed comments on both issues are provided below. My overall assessment is that the manuscript may be suitable for publication in Nature Communications if the authors can clarify these points and confirm the novelty and significance of the work.

“The new results show that, contrary to previous predictions, T_e changes gradually along the chain” This phrase sets the expectation that the new results are revealing a flaw in current thinking, yet it is not clear to me from within the manuscript whether this is really the case. On line 52 we are told that many previous studies yield an abrupt change in T_e at the HESC bend. The most recent of these studies is 1993. Many more recent studies (e.g. Kalnins and Watts, 2009; Lu et al, 2021) don't show an abrupt change at the bend.

The previous T_e estimates along the HESC that are plotted in Supplementary Figure S1 have been compiled from publications during 1970 to 2021. They are mainly based on comparisons of observed geoid, gravity or seismic data to forward models of simple plate flexure along single profiles. They do not include values based on 3D spectral (e.g., admittance, wavelet) methods. The reviewer is correct in pointing out that the Kalnins & Watts (2009) values, which were derived using (moving) windows hundreds of km in width, do not show an abrupt change at the bend. However, as these authors point out, windows of this size tend to smear out spatial variations in T_e . They are therefore unlikely to resolve the abrupt change at the bend as seen in previous forward modeling studies. The Lu et al. (2021) study, in contrast, uses fan wavelet techniques, which potentially have better spatial resolution and do show a change at the bend in the HESC. However, the change is from high values (~35 km) at the Emperor Seamounts to low values (~25 km) at the western end of the Hawaiian Ridge, which is opposite to that shown by the previous forward modeling studies. The Lu et al. (2021) values are therefore difficult to explain since the Emperor Seamounts were formed on much younger oceanic crust than the western end of the HESC.

We have therefore added a discussion of the Kalnins & Watts and Lu et al. results in the Introduction and Motivation section, as suggested by Reviewer.

Further, subsequent text in the 'Results' section states a broad agreement between the new results and some previous studies, so that in the end it is not clear how the new 2D profile model results are providing novel insights.

We hope the reviewer will agree that by adding the range of the previous studies, including the Lu et al reference, and discussing some limitations of the Kalnins & Watts study that we have clarified the uniqueness of our new 3000 estimates of T_e along the whole HESC.

I suggest the authors aim to clarify this point “...weak zones can exist within the interior of a large, otherwise rigid, plate and may, we speculate, facilitate the initiation of intra-oceanic subduction in response, for example, to changes in plate motion and eventually lead to their break-up.” This is an interesting idea, and I had the expectation that the main text would develop it further. However, only the paragraph beginning on line 465 touches on it again. There is a huge literature on subduction initiation that goes completely ignored, and at the least it would be useful for the authors to articulate how they imagine the interpreted weak zones to contribute to existing paradigms - either in a general sense, or making suggestions for specific examples in the recent past (see Gurnis et al, 2004; Lallemand and Arcay, 2021, and many others) where it is at least plausible that weak zones associated with hotspot trails were an important factor.

We are pleased the reviewer considers the weak zone at the HESC a potential site for subduction initiation “an interesting idea”.

We agree with the reviewer that there is huge literature on subduction initiation and so have expanded our discussion to include reference to the works of Gurnis et al. (2004) and Lallemand & Arcay (2021). The latter reference is particularly relevant to this paper as it is one of a number that posits that seamount chains or oceanic plateaus may initiate subduction. One of the examples given in the study is the Afanasy Nikitin Seamount Chain in the Central Indian Ocean where there is evidence of both a weak zone (Paul et al. 1990) and intraplate deformation (Weissel et al. 1980).

We have therefore expanded the discussion on subduction initiation at the end of the Discussion section, adding these references as suggested by the Reviewer.

A related question would be, does the new proposal make testable predictions that would be valuable to other communities. For example, there are many workers studying subduction initiation by mapping and sampling ophiolites, is the implication of this study that we might expect to see certain things in ophiolites interpreted to record some aspects of subduction initiation that is characteristic of weak zones associated with hotspot trails?

We thank the reviewer for this interesting question. There are a number of cases of seamounts and ocean islands that have been described from within ophiolite sequences, for example, in Oman, Turkey, Iran, China and the US. Some of these features have been attributed to mantle plumes, others to intraplate volcanism. The weak zone derived in this paper has a mechanical origin and is based on comparisons of the observed T_e to the calculated T_e based on data from experimental rock mechanics. We do not know the elastic thickness of the seamounts and ocean islands that have been found within ophiolites. However, Watts et al. (2010) pointed out that seamounts and ocean islands that form on weak lithosphere are more likely to be accreted to the forearc, and as a consequence be preserved within ophiolites, while those formed on strong lithosphere are more likely to be carried down into the mantle on the subducting slab. Therefore, we speculate that plume generated seamounts and ocean islands within ophiolites may have either a MORB or OIB geochemical 'fingerprint'.

Although the composition of seamounts and ocean islands is still not well known recent seismic studies along the HESC suggest they comprise a dense core of mafic or ultra-mafic rocks that are draped by extrusive basalts of tholeiitic or alkalic composition, so mafic or ultra-mafic 'pods' might also be expected within ophiolites.

Further related question - what can we say about the nature of weak zones related to volcanic trails in general, as opposed to just the HESC specifically? For example, it should be possible to produce T_e estimates for Louisville and other trails (or use existing ones), considering these could reveal something about whether such weak zones are likely to be common through Earth history or limited to only a subset of trails with Hawaii-like characteristics.

We agree with the reviewer that it would be useful to produce continuous T_e estimates along other seamount chains such as Louisville Ridge. Indeed, we have just begun a study to estimate T_e along the northern and southern segments of this ridge and we hope to be able to report on these results soon in a future paper.

Regarding writing style - I would suggest large sections of text need to be made more accessible for a Nature Communications article, given the target audience should be wider than (for example) GJI or JGR. Certain concepts such as the 'upstream' and 'downstream' flexure are introduced with little context and would require most readers to immediately go and read other sources for background for the text here to be understandable.

We agree with the Reviewer and so have provided better definitions of 'upstream' and 'downstream' flexure where these terms are first mentioned, and have also tried to make the language more accessible throughout the paper.

Regarding uncertainty propagation - I think the authors have done a decent job of considering the many various choices and assumptions that must be made to arrive at T_e estimates in this way. Nonetheless, the final presented values generally appear as a single line of best-fitting T_e versus distance (e.g. Figure 3). We are therefore left to wonder how the magnitude of the variations in this series compares to the uncertainty associated with any or all of the choices at each step. For example, I could imagine that fitting for each profile could yield a range of possible fits that give some kind of confidence region, which together would give an envelope around the best-fitting curve. Of course this would also depend on some assumptions but I think it would be useful to illustrate the magnitude of such envelopes.

We agree with the Reviewer and so have added a new figure to Supplementary, Figure S6, which shows details of the RMS minima for each of the profiles plotted in Figure 2. By assuming a tolerance parameter of 0.1, we estimate our T_e recovery to be accurate to better than ± 2 km.

Other technical comments:

Missing some context, given that we need to understand what constitutes a 'weak zone' in the context of surrounding 'normal' oceanic lithosphere.

I am surprised there is no mention of the work of Lu et al (2021), "What Controls Effective Elastic Thickness of the Lithosphere in the Pacific Ocean", JGR. Full disclosure, I have no particular skin in the game in terms of contrasting interpretations, rather I just want to understand the origins of any contrasting ideas in the current literature. My impression is that the current submission brushes some issues under the carpet. Low T_e at subduction zones seems to be a debated issue, yet the text glosses over this.

As we responded to Reviewer 1, we have added references to both Kalnins & Watts (2007) and Lu et al. (2021) which are spectral methods and therefore are not directly comparable to results based on forward modeling of gravity and seismic data at individual seamounts and ocean islands. There is evidence, for example, that spectral results may smear spatial variations in T_e as shown by Kalnins & Watts in the vicinity of the Hawaiian Ridge where the high T_e associated with the islands extends to the Musician Seamount cluster to the north and the Cook/Jagger Seamount cluster to the south, both of which formed at or near a MOR and so are associated with low, not high, values of T_e .

We are not aware of the debate referred to here by the Reviewer. The results of Hunter & Watts

(2016), for example, show that there is a low T_e at subduction zones, but it is limited to the seaward wall of the trench where swath bathymetry data indicate bend faults and the curvature of flexure is greatest. Elsewhere, in the outer rise region, there is high T_e , which they show are well described by the depth to the 450°C oceanic isotherm based on plate cooling models.

The broader issue to me is that (in furtherance of comments made above) the results presented in the current submission are very much focussed on the HESC without much context (at least in terms of the figures). Consider for example in figure S9, we see plotted some estimated volume fluxes for the HESC with the Wessel (2016) estimates plotted for comparison. This is useful context (even though the flux is a relatively minor issue here. What I would like to see is a much better attempt to illustrate current ideas on T_e within Pacific plate lithosphere (or better, oceanic lithosphere in general) with the new results for the HESC. While all the results appear to be reasonable in themselves, there is a large conceptual leap to the ‘significance for subduction initiation’ which needs to be bridged.

We are pleased the reviewer found the comparison of the volume estimates along the HESC to previous results in Figure S9 useful.

We agree with the Reviewer regarding also showing current ideas on T_e within the Pacific plate and so have added a new figure in Supplementary, Figure S12. The figure compares our Model D results with previous T_e estimates from the French Polynesia region and with other regions of the Pacific Ocean. While our Model results are higher than those from the French Polynesia region (or from seamounts and ocean islands that backtrack to the region) they overlap the estimates from other regions of the Pacific. These estimates are best fit (mean = 2.7 mGal, RMS = 6.1 mGal) by the depth to the 350°C oceanic isotherm, which is similar to the isotherm that describes our new T_e results.

We have commented on this in the caption of Supplementary Figure S12.

Final point, I wonder whether another factor not mentioned relates to the other hotspot trails on the Pacific plate which transect the HESC. For example, older trails transect the HESC (Shatsky, Hess, Musicians), we should expect these to also influence T_e ?

When modeling we applied a mask on the observed bathymetry and gravity data so as to ensure we only considered volcanic loads that could be attributed to the HESC. The Shatsky and Hess rises and the Musician Seamounts, which formed at or near a mid-ocean ridge, are associated with low T_e and are characterized by relatively short wavelength gravity anomalies and flexure. They are therefore likely to have little or no impact on the gravity anomaly or flexure associated with the HESC.

Reviewer 3

After reading the manuscript, I found several aspects that are either insufficiently explained or entirely omitted (specially quantification of parameters through the text), which the authors should address to better convince the reader. Most of these issues are minor, while others are of moderate. Below, I outline several key points:

Abstract:

.....” While low T_e at the Emperor Seamounts and high T_e at the Hawaiian Ridge are expected because of their differences in volcano and plate age,” How are "low" and "high" T_e defined in this context? Please provide quantitative values or ranges—otherwise, the terms remain subjective and open to interpretation....” T_e changes gradually along the chain”.

It is stated that T_e changes gradually from one value to another, but the specific values are not provided. Could you please clarify what X and Y are? Without this information, the statement remains vague.

We have added the ranges of T_e (from Supplementary Figure S1) in the abstract, as suggested by the reviewer.

Introduction and motivation:

Line 44. “thin elastic or viscoelastic plate with uniform”

What exactly is meant by “thin” in this context? While elasticity theory assumes that T_e is much smaller than the plate length, previous studies have reported specific ranges of T_e values for the Hawaiian Ridge and Emperor Seamounts. This information should be clearly explained and acknowledged in the Introduction to provide proper context.

We mean here an elastic beam/plate with a thickness that is small compared to the radius of curvature of flexure. We have pointed this out in the revised paper, as suggested by the Reviewer.

Line 49. “Cretaceous Normal Polarity oceanic crust”

This explanation is clear and appropriate for geoscientists. However, since Nature Communications reaches a broader scientific audience, please consider adding the age range in parentheses—for example, “(121–83 Myr old)” —to provide clearer context for readers from other disciplines.

We have added the age range for the Cretaceous Normal Polarity oceanic crust (83-121 Ma), as suggested by the Reviewer.

Lines 52-54: “While subsequent studies 5-10 have generally confirmed these results, they reveal that T_e changes abruptly at the HESC bend which is not expected from the volcano load and plate ages”

The bend appears more like a kink—an abrupt change in the Pacific Plate’s azimuth. When did this kink occur? According to Wessel & Kroenke (2008, JGR), it happened between 53.4 and 47.9 Ma. This fundamental detail should be clearly stated in the Introduction.

We have added the approximate age (~48 Ma) of the bend and the reference to Wessel & Kroenke (2008), as suggested by the Reviewer.

Figure 1.

- Some isochron values on the oceanic plate are missing. At a minimum, please include isochrons at 50 Myr intervals for better reference.

We have added the missing isochron values to Figure 1, as suggested by the Reviewer. We have not added the 50 Ma age contours, however, as we believe they would make the figure less clear and add little relevant information.

- Additionally, key ages of the Hawaiian Ridge and Emperor Seamounts should be labeled. There are well-established geochemical and geochronological data available for several locations along the HE and ES chains that would help readers understand the age difference between the Pacific Plate and the seamounts. This would effectively illustrate the classical concept of “the age of the plate at the time of volcanic emplacement or loading,” which is critical for interpreting flexure.

We have added key ages to Figure 1 and their references, as suggested by the Reviewer.

Lines 122-123. “for example, with the Hawaiian swell while at the same time maximised the amplitude of the load (Supplementary Figure S2).”

I'm not sure the Hawaiian swell signature can be dismissed a priori.

We agree with the Reviewer that we were perhaps a little too dismissive about removing the effects of swell formation. To address this we modified Supplementary Figure S2 to show the ORS-derived free-air gravity anomaly in addition to the regional bathymetry. The figure shows a close correlation and a ratio of between the regional gravity anomaly and bathymetry at the young end of the HESC of $\sim 21\text{-}27 \text{ mGal km}^{-1}$. This ratio is similar to one found by Watts (1976) and suggests that ORS filtering successfully removes the Hawaiian swell, which has been interpreted to be supported by some form of convection in the mantle.

We have revised the text in the Flexure and Gravity modeling section to reflect this, as suggested by the Reviewer.

McNutt & Bonneville (2000, G-Cubed) and Watts (Chapter 5, Flexure and Isostasy of the Lithosphere, 2nd ed.) highlight the significance of both buried and subsurface loads associated with underplating of anomalously thick crust. This anomalously thick crust is less dense than the surrounding mantle and generates buoyant stresses that contribute to swell formation.

While the volcanic load—with its higher density contrast relative to seawater—likely plays a dominant role in downward flexure, the contribution of the buried load should not be overlooked. Its effect should be quantified, at least along profiles constrained by seismic models that provide density distributions and Moho geometry.

We agree with the Reviewer that in addition to surface loads there is evidence that the oceanic crust upon which seamounts and ocean islands have been emplaced may have also been subject to sub-surface (buried) loads such as those associated with magmatic underplating.

The seismic data in Figure 4, however, show little evidence of magmatic underplating. Rather,

the mantle that underlies the flexed oceanic crust is remarkably homogeneous with regard to its physical properties.

We do not therefore see the need to consider the effects on the free-air gravity field (or the flexure) of magmatic underplating along the HESC.

We note though that the seismic data in Figure 4 reveal another possible sub-surface (buried) load in the form a high velocity, dense, body in the core the volcanic edifice. This is one of the reasons (the other being the need to account for the removal of load during guyot formation) why we used a bathymetry-dependent model based on the seismic P -wave velocity in which density varied from the flank of a seamount or oceanic island to its dense core, rather than using a constant load density

We have added a paragraph on this discussion at the end of the Results section, as suggested by the Reviewer.

Although I expect the results may not vary significantly due to the relatively high effective elastic thickness (T_e) of approximately 20 km along the Hawaiian-Emperor Seamount Chain (HESC), this assumption needs to be tested explicitly along selected profiles and shown in Methods of Supporting Information.

The reviewer is correct in stating that the effect of buried loading may not vary significantly due to the relatively high T_e along the HESC. The problem is testing it since the geometry of the buried load is not well defined in 3D. The only location where it may be tested is on Jimmu Guyot where there is an intersecting seismic dip and strike line. This was carried out at Jimmu in Watts et al. (2021) where the geometry of the buried load was defined by the 6 km/s refractor on the strike and dip lines. We found using a process-oriented approach the gravity effect of the buried load to be strongly localized and the effect on the flexure was to increase the T_e derived from only surface (bathymetric) loading from 14 km to 21 km. We note here that the T_e derived from gravity modeling with a load boost (Model D), which we believe addresses a dense core, was 18 km (Figure 2), which is in better agreement with the 21 km derived previously by Watts et al. (2021).

Figure 3c.

At first glance, this figure appears somewhat unrealistic. Typically, we would expect infill material to accumulate along the flanks of the oceanic ridge, adjacent to the seamount. However, Figure 4 shows that the volcanic edifice extends continuously from the flexed oceanic crust all the way to the core of the seamount. An exception is seen along the V_p model along Line 2, where infill material occupies approximately half of the elevated topography.

We do not believe Figure 3c is unrealistic. This is because it is a strike line through the summits of seamounts and ocean islands that comprise the HESC, not a dip line. The material that infills the moat, which includes volcanoclastic and material derived from mass wasting, is expected to continue into the lowermost flanks of the HESC, but would only be visible on a dip line, not a strike line. The moat infill does not extend across the summits, although the physical properties of this region, which comprise extrusive lavas and clastics that drape intrusive rocks, may well

resemble those of the moat infill. The HESC (light blue shaded region in Figure 3c) comprises the surface ‘driving’ load, which in our calculations is determined directly from the bathymetry. There is infill along the whole HESC, dubbed the root infill, which is the material that has accumulated within the flexure immediately beneath the ‘driving’ load.

This raises the question: how different is the density of the volcanic edifice compared to that of the infill material? If the densities are similar, then Figure 3c could serve as a useful—though simplified—proxy for the system.

The differences in density between moat infill and root infill are given in Figures 2 and Supplementary S6. In general, the moat infill density is somewhat less than the root infill density because the moat infill includes more volcanoclastics and mass wasting products derived from the volcanic edifices while the root infill includes more basaltic lavas associated the initial shield building phase of seamount/ocean island growth. The load density is shown in Supplementary Figure S7.

That said, it is important to recognize the limitations of the model. These include assumptions about uniform density, simplified geometry, and the omission of sediment compaction or lateral variations in lithology. Such simplifications may lead to discrepancies between the model output and the actual structure, particularly in areas where infill material dominates or where crustal flexure varies significantly. Please explain the limitations and uncertainties of the model. It appears that a constant density is assumed for each material body, which may not reflect natural variability.

We agree with the reviewer. We have therefore added a new figure, Supplementary Figure S6, that illustrates the uncertainties in recovering T_e for the 6 selected profiles shown in Figure 2. The uncertainty is better than about ± 2 km, assuming a tolerance parameter of 0.1. We used here a 3D bathymetry-dependent load density model for each seamount and oceanic islands of the HESC in which the load density varies with depth within the volcanic edifice. Then we modeled the flexure using this 3D variable density load, and the gravity effect of the load was calculated. The remaining moat infill and root infill densities may vary along the HESC and are not well determined by the limited seismic data we therefore made them uniform and searched for them along each of the profiles. Moreover, we used a ‘load boost’ to allow consideration of the mass missing that may have contributed to the gravity anomaly and flexure at the time of volcano loading.

Figure 4. Please clarify: were the dotted lines calculated using a constant or variable effective elastic thickness (T_e)? What range of T_e values best fits the data in panels (a), (b), and (c)? This information should be clearly specified in the figure caption.

The white dashed lines in Figure 4 shows the calculated flexure assuming a constant T_e as determined from the 2D gravity modeling along each of the 3000 profiles. We are not attempting to derive the T_e that best fit the seismic data (this has already been undertaken in the papers by Xu et al., Dunn et al. and MacGregor et al..

The T_e and density values used to construct the flexure in Figure 4 are listed in Supplementary Table S1 and the T_e values in this table have been added to the caption of Figure 4, as suggested

by the Reviewer.

Figures 5 and 6. It is surprising that the “yield T_e ” is approximately twice as large as in the purely elastic model, even though $T_e(x)$ is allowed to vary and decrease in regions of high plate curvature in the elastic model. The yield T_e —or more precisely, the yield strength envelope (YSE)—is known to depend on plate curvature, as shown in Figure 5, but it is also highly sensitive to several parameters: the strain rate ($\dot{\epsilon}$), activation energy (Q), the stress exponent in the ductile flow law (n), constant of the material (A), the friction coefficient (μ), the fluid pore pressure ratio (λ), and the thermal structure ($T(z)$).

Our approach has been to take the classic experimental results of Goetze and Goetze & Evans and compare the ‘yield T_e ’ implied by their rheology to the T_e derived from gravity modeling along the HESC. We chose this rheology because it has been widely used in flexure studies and Hunter & Watts (2016) suggested in their inversion “A1” that it worked well at subduction zones. We found, however, using the observed curvatures of flexure that the rheology was too strong for the HESC and so the question was could we change the parameters in this rheology and fit the observed T_e ? To do this we tested the dependence of the ‘Yield T_e ’ Goetze and Goetze & Evans rheology on activation energy (Q_p , Q_d), constant of the material (A_p), friction coefficient (μ_f), and thermal structure $T(z)$. We chose these test parameters above the others because they had been varied in previous studies (e.g. Zhong & Watts) and are probably the most uncertain of all parameters in the laboratory-derived rheological laws.

An explicit expression for the yield deviatoric stress should be included in the Supporting Information or Methods section. There are various formulations beyond the exponential laws, and the specific choice is critical for deriving a higher “yield T_e ” compared to the purely elastic model.

We have included the expressions for the yield stress that we used in the new Supplementary Table S2, as suggested by the Reviewer.

Later, the authors mention that “...the brittle and ductile flow laws are too strong to explain the observed T_e at the HESC...”. At this point in the manuscript, they should explicitly state the values of the critical parameters used ($\dot{\epsilon}$, A , Q , μ , n , and λ), rather than waiting until Figure 6 to discuss the calibration of μ , n , Q , and $T(z)$ to match the observations.

We have added the values of $\dot{\epsilon}$, A_p , Q_p and Q_d , and μ to the first line where it is stated that the rheological laws are too strong, as suggested by the Reviewer. Other parameters are listed in the new Supplementary Table S2.

A few questions arise from this analysis:

- Why was the thermal age modified? It is generally well constrained, except in specific regions affected by plume-related thermal rejuvenation.

The thermal age was modified because it has recently been proposed (e.g. Pleus et al. 2020, GJI; Douglas et al. 2025, JGR) that the Pacific lithosphere may have been rejuvenated by the thermal effects of the Hawaiian hotspot or magma assisted flexure.

- What is the impact of decreasing $\dot{\epsilon}$ from 10^{-14} to 10^{-15} or 10^{-16} s^{-1} ? In fact, Hunter & Watts (2016) used $\dot{\epsilon} = 10^{-16} \text{ s}^{-1}$. What strain rate was used in Figure 5c for the “strong” Pacific Plate?

We selected $\dot{\epsilon} = 10^{-14} \text{ s}^{-1}$ based on consideration of a) the duration of time it takes for a ‘typical’ seamount or ocean island to form and b) the relaxation that takes place as the thickness of the lithosphere that supports a volcanic load decreases from its short-term (seismic?) thickness to its long-term elastic thickness. For example, an Emperor seamount that took 2.7 Myr to form on 50 Ma oceanic lithosphere with a seismic thickness of 95 km and a T_e of 14 km would be associated with a strain rate of $\dot{\epsilon} = 10^{-14} \text{ s}^{-1}$. Decreasing the strain rate to 10^{-15} s^{-1} or 10^{-16} s^{-1} for the same thermal age, seismic thickness and T_e , however, would require a duration of seamount formation of 27 and 270 Myr respectively which are a factor of 10 and 100 times respectively longer than normally considered. Similar results were obtained for a seamount in the Hawaiian Ridge.

Irrespective, we have tested the effects of decreasing the strain rate to the values used by Goetze (10^{-15} s^{-1}) and & Watts (10^{-16} s^{-1}). We found that decreasing the strain rate weakens the plate. However, the effect is small and in the case of an Emperor seamount the result would be a reduction in the ‘Yield T_e ’ by 1.5-3 km and 3-6 km for a strain rate of 10^{-15} s^{-1} and 10^{-16} s^{-1} respectively. Similar results were obtained for a seamount in the Hawaiian Ridge.

We have made these points clearer in the revised text and Supplementary Table S3.

The T_e in Figure 5c for the “strong” Pacific plate is based on the depth to the 671-714°C oceanic isotherms which Hunter & Watts (2016) showed describe well the observed T_e oceanward of the weak zone at deep-sea trenches. It is not dependent on a strain rate.

We have made this clearer in the revised caption to Figure 5.

- What value of the fluid pore pressure ratio (β) was applied in the brittle portion of the YSE calculations?

We assumed fluid-pore pressure = 0 which is the case for ‘dry’ conditions as used by Goetze.

- In Figure 6 shows the variation of Q are shown, but it is unclear what value of A was used in the calculations.

The value of the pre-exponential, A_p , used in Figure 6b is $7.0 \times 10^{-14} \text{ Pa}\cdot\text{s}^{-1}$ and is from Goetze. Note that the variation of A_p in the figure show the factors (e.g. 1, 10^4 , 10^8 and 10^{12}) by which this value has been increased. We made this clearer in the Figure 6b caption, as suggested by the Reviewer.

The geological cause of the apparent plate weakening remains unclear. From the perspective of the YSE, is this weakening primarily attributed to changes in A and n , or to a lower $\dot{\epsilon}$? Which factor is more likely responsible?

We have shown in this paper that the rheological laws of Goetze and Evans & Goetze – and

probably other LPT laws - are too strong to explain the observed T_e variation along the HESC. The Pacific plate underlying the HESC is therefore weaker than expected by these laboratory-derived rheological laws. The cause of this weakness is not clear, but we believe it to be a mechanical, not a thermal, one. We present evidence in Figure 6 that plausible increases in the pre-exponential constant and decreases in the activation energy could account for the degree of weakness. Decreases in frictional coefficient or thermal age at the time of loading might also explain the weakness but they imply parameter values such as <0.075 and 67-83%, respectively, to explain our observations. We do not believe that other factors that might weaken the lithosphere such as an increase in the pore-fluid pressure (say to hydrostatic conditions) or decreases in the strain rate would be significant enough to explain the weakness. Therefore, we conclude that increases in the pre-exponential constant, decreases in the activation energy or some combination of these parameters best account for the observed T_e variation.

Finally, it would be very helpful to include a brief sentence indicating the typical range of plate curvature values in outer-rise settings, and how these compare to those associated with flexure due to seamount loading.

The plate curvature values in circum-Pacific outer-rise settings are given in figure 6a of Hunter & Watts (2016) and are in the range $4.5 \times 10^{-7} \text{ m}^{-1}$ to $11.0 \times 10^{-7} \text{ m}^{-1}$. Comparisons of these values show them to be of the order or lower than the maximum positive curvatures along the Emperor Seamounts and generally higher than the maximum positive curvatures along the Hawaiian Ridge.

We have added the range of outer rise curvatures to Figure 5a so that comparisons can be made with the seamount and ocean island curvatures, as suggested by the Reviewer.

The last sentence "... we speculate, facilitate intra-oceanic subduction in response, for example, to changes in plate motion and eventually lead to their break-up" is identical to the last sentence of the Summary/Abstract.

We have modified the sentence at the end of the Discussion slightly, as suggested by the Reviewer.

Although this is a speculation, plate breakup is typically accompanied by thermal upwellings and extensional stresses. Could you elaborate a bit more on this speculation and clarify the underlying mechanisms you envision?

We have expanded the discussion on this speculation at the end of the Discussion section, as suggested by the Reviewer.

Comment: I agree with the authors that the weakening observed at both the HESC and CPTOR is primarily a consequence of mechanical loading rather than thermal processes. This suggests a role for spatial variations in the parameters governing both brittle and ductile deformation, as has been proposed.

We are pleased the Reviewer agrees with the cause of the weakening at both the HESC and CPTOR.

Videos.

The videos are very good and illustrative. However, the physical units on both the horizontal and vertical axes are missing.

We are pleased the Reviewer found the videos instructive and have added the physical units on the vertical axes, as suggested by the Reviewer.

Editor

You will see that while the [reviewers] are interested in your study, they also raise concerns. These include the discussion of previous results, the implications of subduction initiation, the influence of density variations within the mantle lithosphere and the limitations and uncertainties of the model.

We believe that we have now addressed all the concerns in our response to the Reviewers. In particular, the discussion of previous results have been addressed in the “Introduction and Motivation” section in response to Reviewer 2, the implications of subduction initiation have been fully addressed in an expanded “Discussion” section in response to Reviewer 2, the influence of density variations within the mantle lithosphere have been addressed in the “Flexure and Gravity modeling” section in response to Reviewer 1 and the limitations and uncertainties of the model have been addressed in the “Results” section and in Supplementary Figure S6 in the response to Reviewer 3.

We hope these responses to Reviewers 1-3 and the Editor are satisfactory and we look forward to hearing from you.

NCOMMS-25-18554A

28 August 2025

In the following, black font shows the comment of the Reviewers and red font our response.

Reviewer 1

The authors have well responded to my comments. I think the manuscript is more accessible to the general audience now. I have no further comments.

We are pleased the reviewer found we “have responded well to (his/her) comments” and that (he/her) has no further comments.

Reviewer 2

I have not much to add based on the revised manuscript. The authors have made a good attempt to respond to comments, just a few new issues:

We are pleased the reviewer found we “have made a good attempt to respond to (his/her) comments”.

Figure 5 and related text, paragraph 417: I would find it useful to make a more explicit connection between the text and the two alternative green curves, provide a basic explanation for what the 'weak zone' and 'strong zone' curves are and why they are significant here.

Line 437 - consider shortening sentence.

We have shortened the sentence by breaking it into two, as suggested by the reviewer.

Line 511 - consider shortening sentence.

We have shortened the sentence by breaking it into two, as suggested by the reviewer.

Line 529: "...seamount chains or oceanic plateaus may initiate subduction..."

Consider the wording here, I suspect you mean to convey that these features are sites of weakness that may control where subduction initiates (driven by forcing from elsewhere). Rather than being the driver themselves.

We have changed the wording, as suggested by the reviewer.

Reviewer 3

The authors have addressed my suggestions, and the manuscript is now much improved. I have no further requests.

We are pleased the reviewer found that we “addressed (his) suggestions, and the manuscript is now much improved”.

We hope these responses to Reviewers 1-3 are satisfactory and we look forward to hearing from you.